# Molecular basis for assembly of the shieldin complex and its implications for NHEJ

Ling Liang [1,2,5✉], Jiawen Feng [1,5], Peng Zuo [1], Juan Yang [1,2], Yishuo Lu [1,3] & Yuxin Yin [1,3,4✉]

Shieldin, including SHLD1, SHLD2, SHLD3 and REV7, functions as a bridge linking 53BP1-RIF1 and single-strand DNA to suppress the DNA termini nucleolytic resection during non-homologous end joining (NHEJ). However, the mechanism of shieldin assembly remains unclear. Here we present the crystal structure of the SHLD3-REV7-SHLD2 ternary complex and reveal an unexpected C (closed)-REV7-O (open)-REV7 conformational dimer mediated by SHLD3. We show that SHLD2 interacts with O-REV7 and the N-terminus of SHLD3 by forming β sheet sandwich. Disruption of the REV7 conformational dimer abolishes the assembly of shieldin and impairs NHEJ efficiency. The conserved FXPWFP motif of SHLD3 binds to C-REV7 and blocks its binding to REV1, which excludes shieldin from the REV1/Pol ζ translesion synthesis (TLS) complex. Our study reveals the molecular architecture of shieldin assembly, elucidates the structural basis of the REV7 conformational dimer, and provides mechanistic insight into orchestration between TLS and NHEJ.

[1] Institute of Systems Biomedicine, Department of Pathology, Beijing Key Laboratory of Tumor Systems Biology, School of Basic Medical Sciences, Peking University Health Science Center, Beijing 100191, China. [2] Department of Biochemistry and Biophysics, School of Basic Medical Sciences, Peking University Health Science Center, Beijing 100191, China. [3] Peking-Tsinghua Center for Life Sciences, Peking University, Beijing 100871, China. [4] Institute of Precision Medicine, Peking University Shenzhen Hospital, Shenzhen 518036, China. [5] These authors contributed equally: Ling Liang, Jiawen Feng. ✉email: liangling@hsc.pku.edu.cn; yinyuxin@hsc.pku.edu.cn

Non-homologous end joining (NHEJ) is an important physiological process involved in class-switch recombination (CSR), fusion of unprotected telomeres and repair of intrachromosomal breaks[1]. Shielding DNA ends is quite important for triggering NHEJ in DNA double-strand breaks (DSBs)[2]. It is known that shieldin, which is composed of SHLD1, SHLD2, SHLD3 and REV7, functions as a downstream effector of 53BP1-RIF1 to suppress the DNA termini nucleolytic resection by binding to single-strand DNA (ssDNA) during NHEJ[1,3–9].

In the four-subunit protein complex, REV7 was the first component to be reported as a counteractor of DNA DSB resection and an important regulator in choice of DSB repair pathway in 2015[10,11]. REV7, also known as MAD2B and MAD2L2, is a conserved protein of the HORMA domain (named after the Hop1, REV7 and Mad2 proteins) family and undergoes an open (O)-to-closed (C) transition when partner proteins bind to the safety-belt[12]. REV7 was first found to be a subunit of Pol ζ, a translesion synthesis (TLS) polymerase that enables replication of damaged DNA[12–14]. Pol ζ is composed of REV3, REV7, PolD2 and PolD3, in which REV3 is the catalytic subunit and REV7 acts as a multitasking scaffolding protein[13,15,16]. Human REV3 (hREV3, hereafter called REV3) has over 3,000 residues and is twice as large as its homolog in yeast. Except for the relatively conserved N-terminal 250 amino acids and C-terminal 800 amino acids, which are homologous to Pol α, δ, and ε, but without 3′–5′ exonuclease activity, REV3 has a positively charged domain (PCD, amino acids 1042–1251) for DNA binding and two REV7-binding motifs (RBMs). The two RBMs in REV3 (amino acids 1847–2021, hereafter called REV3 (1847–2021)) are characterized as PXXXpP motif and both RBM$_1$ (amino acids 1847-1906) and RBM$_2$ (amino acids 1977–2021) bind to REV7 underneath the safety belt loop as other reported REV7 binding proteins, such as human RAN, ELK-1, chromosome alignment–maintaining phosphoprotein (CAMP) and *shigella* IpaB[17–22]. To date, all reported crystal structures of REV7 complexed with its partners adopt a similar closed conformation as a monomer with RBMs bound underneath the safety-belt loop[12,17,21–25].

Since SHLD3 has two REV3-like RBM motifs, it is supposed that SHLD3 interacts with the C-terminal safety-belt of REV7 in a similar manner[6]. However, whether these two RBMs interact with REV7 in the same way as REV3 is uncertain. On the other hand, although it is known that the N terminus of SHLD2 (amino acids 1–60, hereafter called SHLD2(1–60)) is sufficient for its interaction with upstream molecules SHLD3 and REV7[1,4], neither SHLD3 nor REV7 interacts with SHLD2 solely[4,6], the details of their interactions needs to be further explored to understand how shieldin is assembled.

In this study, we solved the crystal structure of the SHLD3-REV7-SHLD2 ternary complex. We demonstrate that SHLD3 binds to REV7 in a completely different way from that of other REV7 binding proteins. Two copies of REV7 bind to SHLD3, and REV7 adopts two conformations with different topologies, closed and open states. O-REV7 is essential for the interaction between SHLD3-REV7 sub-complex and SHLD2 by forming β sheet sandwich occupying the position of the safety belt. Further evidence shows the conformational dimer precludes the binding of C-REV7 to REV1 C-terminal domain (CTD) and may act as a platform to interact with other REV7 binding proteins, such as REV3. Taken together, our work illustrates how REV7 interacts with other components of shieldin through its conformational change, and reveals NHEJ and TLS are mutually exclusive events coordinated by REV7.

## Results

### Overall structure of the SHLD3-REV7-SHLD2 complex. Shieldin complex is composed of REV7 and three newly characterized

proteins SHLD1, SHLD2 and SHLD3[1,4,6]. REV7 contains a HORMA domain that usually acts as an adaptor to recruit other proteins. SHLD2 contains an N-terminal REV7 interacting motif (RIM) and a C-terminal OB fold domain that resembles RPA70, which is connected by a predicated disordered linker[1,4]. SHLD3 contains two putative REV7-binding motifs (RBM) in N terminus and an EIF4E-like motif in C terminus[6] (Fig. 1a).

To elucidate how shieldin complex is assembled, we constructed full length REV7 and the N-terminal REV7-binding domains of both SHLD3 and SHLD2 (designated as SHLD3(1-64) and SHLD2(1-52)), co-purified REV7-SHLD3(1-64)-SHLD2(1-52), and solved the crystal structure of the complex to 3.5 Å resolution (Fig. 1a and Table 1). The high-quality electron density map made unambiguous building of the structure model possible (Supplementary Fig. 1). Unexpectedly, the crystal structure shows that SHLD3-SHLD2 binds two REV7 molecules, one in its closed (C) state and the other in an open (O) conformation, which are characterized by the conformation of the safety belt, thereafter we name it as C-REV7-O-REV7 conformational dimer (Fig. 1b). Compared with C-REV7, the αC, β1, β2 and β7 of O-REV7 show obvious rearrangement despite the safety belt (Fig. 1c, d). As a consensus REV7-binding motif PXXXpP, RBM$_2$ of SHLD3 interacts with C-REV7 in a canonical manner as RBMs of REV3 (Supplementary Fig. 2). However, the recognition mechanism between SHLD3 and O-REV7 is completely different (Fig. 1b). Indeed, RBM$_1$ ([11]PCESDP[16]) of SHLD3 is not a consensus REV7-binding motif PXXXpP, and this is in accordance with the observation that only Pro11 is conserved across species as Pro53 and Pro58 while Pro16 is not (Fig. 1e). Instead, SHLD2 helps SHLD3 interact with O-REV7 and acts as a bolt to lock SHLD3 and O-REV7 tightly, which we will discuss in detail later (Fig. 1b).

**The interface between C-REV7 and O-REV7.** In our crystal structure, REV7 forms a conformational dimer which resembles C-Mad2-O-Mad2 but is different[26]. Compared with the known C-Mad2-O-Mad2 conformational dimer, the obvious difference is that SHLD3 acts as a bridge to link C-REV7 and O-REV7 (Supplementary Fig. 3a). Similar to C-Mad2-O-Mad2, the REV7 conformational dimer uses the conventional HORMA interface centered around αC helix to dimerize and the dimer interface is mainly connected by hydrogen bonds and electrostatic interactions with a buried area of 1670 Å$^2$ (Fig. 2a, b). Several residues make key interactions including Glu35, Lys44, Arg124, Lys129, Asp134 and Lys190 (Fig. 2a). Discriminatively, Arg124 and Asp134 of both REV7 molecules are located at the interface and contributes to the interaction, while Glu35, Lys44 of C-REV7 and Lys129, Lys190 of O-REV7 diverge from the interface and have no effect on the interaction (Fig. 2a, c). When complexed with RBM$_2$ of SHLD3, REV7 forms a closed conformation (C-REV7), while REV7 adopts an open conformation (O-REV7) without partners. Size-exclusion chromatography with multi-angle light scattering (SEC-MALS) shows the samples of REV7$^{WT}$-SHLD3 (WT: wild type) are quite heterogeneous and are proposed to be composed of C-REV7-O-REV7-SHLD3(1–82) and C-REV7-SHLD3(1–82), which were verified later (Supplementary Fig. 3b and Supplementary Table 1). Substitution of Glu35, Lys44, Arg124, Lys129 or Lys190 of REV7 at the dimer interface with an alanine residue abolishes formation of SHLD3 mediated REV7 conformational dimer and forms homogeneous C-REV7-SHLD3 (1–82) heterodimer (Supplementary Fig. 3c–h and Supplementary Table 1). Consistent with the structural data, isothermal titration calorimetry (ITC) experiments showed that REV7$^{E35A}$-SHLD3(1–82) or REV7$^{K44A}$-SHLD3(1–82) binds to REV7$^{K129A}$ (O-REV7) with a binding affinity of about 280 ± 20 nM or 110 ± 10 nM, while REV7$^{K129A}$-SHLD3(1–82) or REV7$^{K190A}$-SHLD3

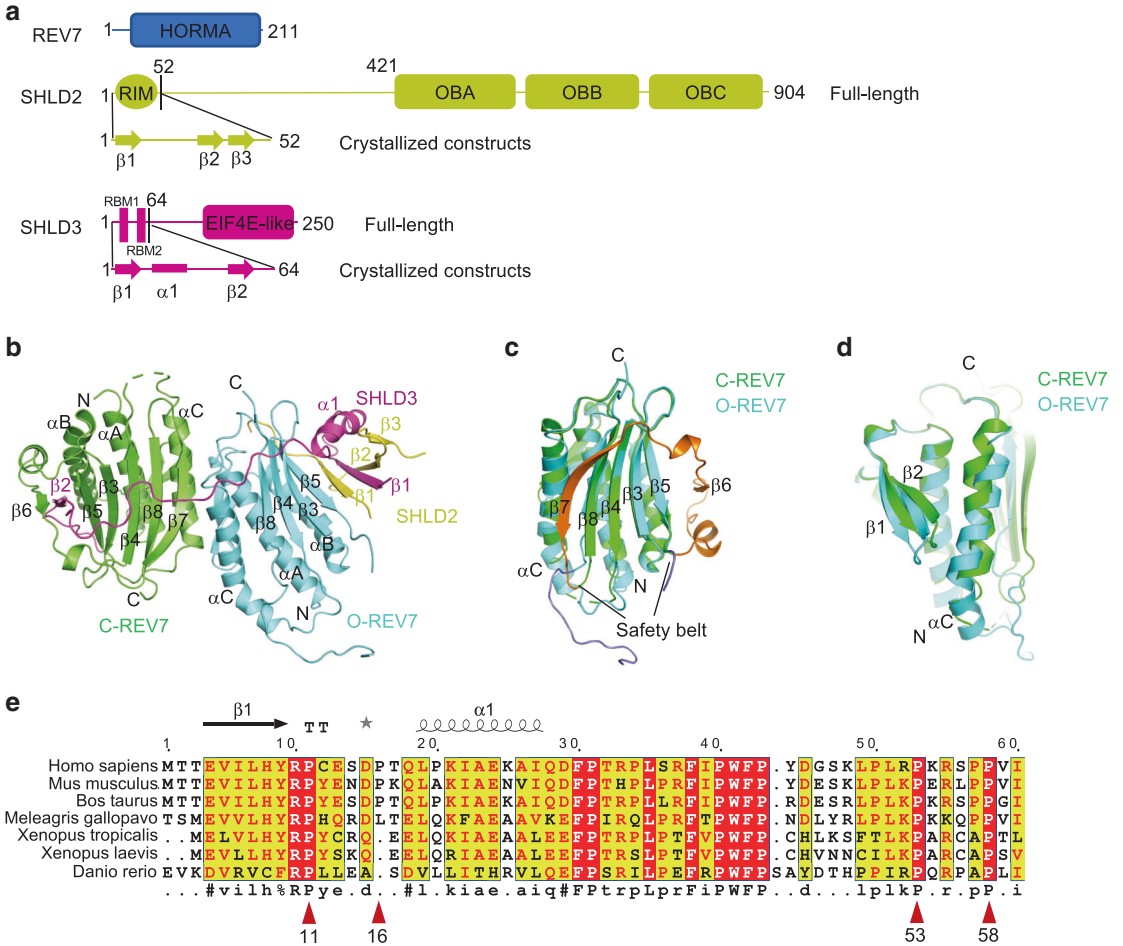

**Fig. 1 Overall structure of the SHLD3-REV7-SHLD2 ternary complex. a** Schematic representation showing the organization of REV7, SHLD2 and SHLD3. The secondary structures of the truncations of SHLD2 and SHLD3 used for crystallization are shown in detail, arrow indicates β strand and rectangle represents α helix. HORMA, a domain named after the Hop1, REV7 and Mad2 proteins; RIM, REV7 interacting motif; RBM, REV7 binding motif, characterized as PXXXpP. **b** Structure of the SHLD3-C-REV7-O-REV7-SHLD2 complex. Two REV7 molecules are differentially colored to indicate their different states, C-REV7 is shown in green and O-REV7 is shown in cyan. The secondary structures of C-REV7, O-REV7, SHLD2 and SHLD3 are labeled. Disordered loop is shown as dashed lines. **c** Structural alignment of C-REV7 and O-REV7. The regions between two lines represent the safety belt and are colored in orange (C-REV7) and slate (O-REV7). **d** Structural alignment of C-REV7 and O-REV7 viewed in another side, which shows the different positions of αC, β1, β2 between C-REV7 and O-REV7. **e** Sequence alignment of SHLD3(1–60) across species. The highly conserved residues are shown in red background. The prolines which are proposed as the conserved PXXXpP motif are indicated with red triangle and numbered.

(1–82) hardly binds to REV7$^{K129A}$ (Fig. 2d–g). This is because Lys129 and Lys190 of C-REV7 locate at the asymmetric interface of C-REV7-O-REV7, and contribute to the interaction, while Glu35, Lys44 of C-REV7 diverge from the interface and have no effect on the interaction. Collectively, REV7 utilizes the conventional HORMA interface to form an asymmetric conformational dimer.

**SHLD3 enhances the interaction between C-REV7 and O-REV7.** SHLD3 acts as a bridge to link C-REV7 and O-REV7, we wondered whether SHLD3 further strengthens the REV7 conformational dimer. Although RBM$_2$ of SHLD3 interacts with C-REV7 in a canonical manner, the structure also shows that $^{38}$FXPWFP$^{43}$ makes extensive interactions with C-REV7 despite RBM$_2$, and these residues are all highly conserved across species, which indicates FXPWFP may be an unrevealed C-REV7 binding motif and exerts important functions (Fig. 3a and Fig. 1e). The conserved FXPWFP motif is composed of hydrophobic residues and binds a relatively hydrophobic surface of C-REV7 (Fig. 3a). Despite the hydrophobic interactions, many hydrogen bonds also contribute to the interaction. In brief, REV7 residues Lys82,

Glu101, Gln200 and Tyr202 form hydrogen bonds with backbone of the FXPWFP motif and Gln200 of REV7 also forms a hydrogen bond with the side chain of Trp41$^{SHLD3}$ (Fig. 3b). Moreover, with Phe38$^{SHLD3}$ in the center, residues Phe38, Pro40 of SHLD3, residues Leu186, Pro188, Tyr202 of C-REV7 and residues Val132, Ala135, Val136 of O-REV7 make extensive hydrophobic interactions (Fig. 3b).

Since Phe38$^{SHLD3}$ locates at the interface between C-REV7 and O-REV7, we suspect it also contributes to conformational dimer formation. We therefore performed ITC to test Phe38$^{SHLD3}$ mutation in interaction between C-REV7 and O-REV7. As previous results showed a single mutation of key residues in the dimer interface forms homogeneous C-REV7-SHLD3(1–82) heterodimer and to keep the experiments consistent, hereafter we use REV7$^{K44A}$-SHLD3 as C-REV7 and REV7$^{K129A}$ as O-REV7. As expected, REV7$^{K44A}$-SHLD3(1–82)$^{F38A}$ binds to REV7$^{K129A}$ with a much lower binding affinity as compared with REV7$^{K44A}$-SHLD3 (1–82) (1.8 ± 0.09 μM versus 0.11 ± 0.01 μM) (Fig. 3c and Fig. 2e). Moreover, as shown in Fig. 3d and Fig. 3e, the binding affinity between REV7$^{K44A}$-SHLD3(38–82) or REV7$^{K44A}$-SHLD3(28-82) and REV7$^{K129A}$ is 0.47 ± 0.17 μM and 0.20 ± 0.01 μM, respectively,

**Table 1 Data collection and structure refinement statistics.**

| PDB | 6KTO |
|---|---|
| Data collection | |
| Wavelength, Å | 0.97853 |
| Space group | P6122 |
| Cell dimensions | |
| a, b, c (Å) | 93.843, 93.843, 325.377 |
| α, β, γ (°) | 90, 90, 120 |
| Resolution (Å) | 50-3.45(3.72-3.45)* |
| $R_{pim}$ (%) | 5.8(45.2) |
| $I / \delta I$ | 12.75(2.0) |
| Completeness (%) | 100(100) |
| Redundancy | 13.0(12.4) |
| Refinement | |
| Resolution (Å) | 46.441-3.45 |
| No. reflections | 11237 |
| $R_{work} / R_{free}$ | 0.242/0.269 |
| No. atoms | |
| Protein | 3710 |
| B-factors | |
| Protein | 47.65 |
| R.m.s. deviations | |
| Bond lengths (Å) | 0.013 |
| Bond angles (°) | 1.494 |
| Ramachandran plot | |
| Favored, % | 93.89 |
| Allowed, % | 6.11 |
| Disallowed, % | 0 |

*Values in parentheses are for highest-resolution shell. One crystal was used for the data set.

lower than REV7$^{K44A}$-SHLD3(1–82) (0.11 ± 0.01 µM), indicating that residues 28-37 and residues 1–27 of SHLD3 also contribute to the physical interaction between C-REV7-SHLD3 and O-REV7 but with limited impact. Furthermore, the binding affinity between REV7$^{K44A}$-SHLD3(45–82) and REV7$^{K129A}$ is 4.1 ± 0.19 µM, which is just a little lower than REV7$^{K44A}$-SHLD3(1–82)$^{FIPWFP/AIAAAA}$ (hereafter short as REV7$^{K44A}$-SHLD3(1–82)$^{5A}$) (3.3 ± 1.1 µM), but is much lower than REV7$^{K44A}$-SHLD3(38–82) (0.47 ± 0.17 µM) (Fig. 3d and Supplementary Fig. 4a, b). This confirms that the highly conserved FXPWFP motif of SHLD3 significantly enhances the binding affinity between C-REV7 and O-REV7 while SHLD3 (1–37) contributes limited impact.

**The FXPWFP motif blocks the REV7-REV1 interaction.** It is reported that residues Leu186, Gln200 and Tyr202 of C-REV7 that contributes its binding to SHLD3 are significantly involved in REV1 binding[23–25]. We superimposed the structures of the C-REV7-SHLD3 complex and the REV3-REV7-REV1 complex, which shows the FXPWFP motif absolutely occupies the binding site of C-REV7 for REV1 (Fig. 4a). Next, we performed ITC experiment to investigate whether the FXPWFP motif in SHLD3 preclude the REV1 binding to REV7 in solution. An initial attempt to purify REV1 CTD was unsuccessful because the instability of the REV1 CTD in solution. As previous report[27], we fused POL κ RIR (REV1-interacting region, residues 562–577) peptide to the REV1 CTD (residues 1140–1251) N-terminus and greatly improved the stability. Hereafter we refer this chimeric REV1 CTD as cREV1 CTD. cREV1 CTD fails to interact with REV7$^{K44A}$-SHLD3(1–82) and REV7$^{K44A}$-SHLD3(38–82) (Fig. 4b, c), however, it still binds to REV7$^{K44A}$-SHLD3(1–82)$^{5A}$ and REV7$^{K44A}$-SHLD3(45–82) with binding affinities of 1.1 ± 0.1 µM and 1.4 ± 0.25 µM (Fig. 4d, e), which is approximate to the binding affinity between cREV1 CTD and a representative C-REV7, REV7$^{WT}$-REV3(1871–2021) (Fig. 4f), suggesting the

FXPWFP motif of SHLD3 disrupts the interaction between REV1 and C-REV7. Moreover, REV7$^{K44A}$-SHLD3(1–82)$^{F38A}$ shows a lower binding affinity to cREV1 CTD than that of REV7$^{K44A}$-SHLD3(1–82)$^{5A}$ (Fig. 4g versus 4d, 6.5 ± 1.2 µM versus 1.1 ± 0.1 µM), which indicates residues other than Phe38$^{SHLD3}$ in the FXPWFP motif still contribute to the binding between SHLD3 and C-REV7. These results indicate that the highly conserved FXPWFP motif of SHLD3 inhibits the REV1-REV7 interaction.

**C-REV7-O-REV7 is essential for the recruitment of SHLD2.** To determine whether SHLD3 mediated REV7 conformational dimer is essential for binding to SHLD2, we performed gel filtration assay. Because SHLD2(1–60) cannot be expressed solely in a soluble state, we fused a MBP tag to its N terminus, which is designated as MBP-SHLD2(1–60). As shown in Fig. 5a and Fig. 5b, MBP-SHLD2(1–60) forms a stable complex with REV7$^{WT}$-SHLD3(1–82), but not with mutant REV7$^{R124A}$-SHLD3 (1–82) that fails to form the conformational dimer, indicating that C-REV7-SHLD3(1–82) without O-REV7 fails to interact with MBP-SHLD2(1–60). Although MBP-SHLD2(1–60) is excessive, only a part of REV7$^{WT}$-SHLD3(1–82) forms stable complexes with MBP-SHLD2(1–60) (Fig. 5a, labeled in black in both left and right panels, 12.5 ml–13 ml). This is in consistence with our previous SEC-MALS data, which shows that our recombinant expressed REV7$^{WT}$-SHLD3(1–82) is not homogeneous and is composed of C-REV7-O-REV7-SHLD3(1–82) and C-REV7-SHLD3(1–82) (Supplementary Fig. 3b). This can be confirmed by the observation that the ratio of REV7 to SHLD3 (1–82) is higher in the peak eluted at 12.5–13 ml than in the peak eluted at 14.5–15 ml (Fig. 5a, 9.6/1.0 versus 8.0/2.5, the gray values are labeled in black box in right panel). As REV7$^{R124A}$-SHLD3(1–82) solely does, REV7$^{E35A}$-SHLD3(1–82) solely fails to form conformational dimer and thus fails to interact with MBP-SHLD2(1–60) (Fig. 5c, pink line). As expected, the reconstituted conformational dimer utilizing asymmetric REV7$^{E35A}$ or REV7$^{K44A}$ and REV7$^{K129A}$ mutations form stable complexes with MBP-SHLD2(1–60) (Fig. 5c, black and red line), while REV7$^{K129A}$-SHLD3(1–82) and REV7$^{K129A}$ reconstituted much weaker conformational dimer form fewer complexes with MBP-SHLD2(1–60) (Fig. 5c, blue line). This was further confirmed by ITC experiments, in which the binding affinity of REV7$^{E35A}$-SHLD3(1–82) and REV7$^{K129A}$-SHLD3(1–82) to REV7$^{K129A}$ in the context of MBP-SHLD2(1–60) is 60 ± 19 nM and 410 ± 150 nM, respectively (Supplementary Fig. 5a, b). Moreover, REV7$^{E35A}$-SHLD3(1–82) and other mutants that fail to form the conformational dimer also show only weak interaction with MBP-SHLD2(1–60) in the pulldown assay (Fig. 5d, lanes 9–12). In contrast, in presence of REV7$^{K129A}$, MBP-SHLD2(1–60) shows stronger interaction with REV7$^{E35A/K44A}$-SHLD3(1–82) that form reconstituted conformational dimer utilizing asymmetric REV7$^{E35A/K44A}$ and REV7$^{K129A}$ mutations than with REV7$^{K129A/K190A}$-SHLD3(1–82) that fails to form the conformational dimer with REV7$^{K129A}$ (Fig. 5d, lane 13 and 14 versus lane 15 and 16). Taken together, we demonstrate that C-REV7-SHLD3(1–82) without O-REV7 fails to interact with MBP-SHLD2(1–60), and mutation interfering the C-REV7-O-REV7 interface impairs the assembly of the SHLD3-REV7-SHLD2 complex.

REV7$^{K190A}$-SHLD3(1–82) fails to interact with O-REV7 as we previously showed (Fig. 2g), which is essential for the assembly of shieldin, we determined whether it interferes with NHEJ. As expected, overexpression of REV7$^{K190A}$ but not REV7$^{K44A}$ reduced NHEJ efficiency in a dominant-negative manner (Fig. 5e). This is because REV7$^{K44A}$-SHLD3 still recruits endogenous REV7$^{WT}$ to assemble the shieldin complex while REV7$^{K190A}$-SHLD3 precludes

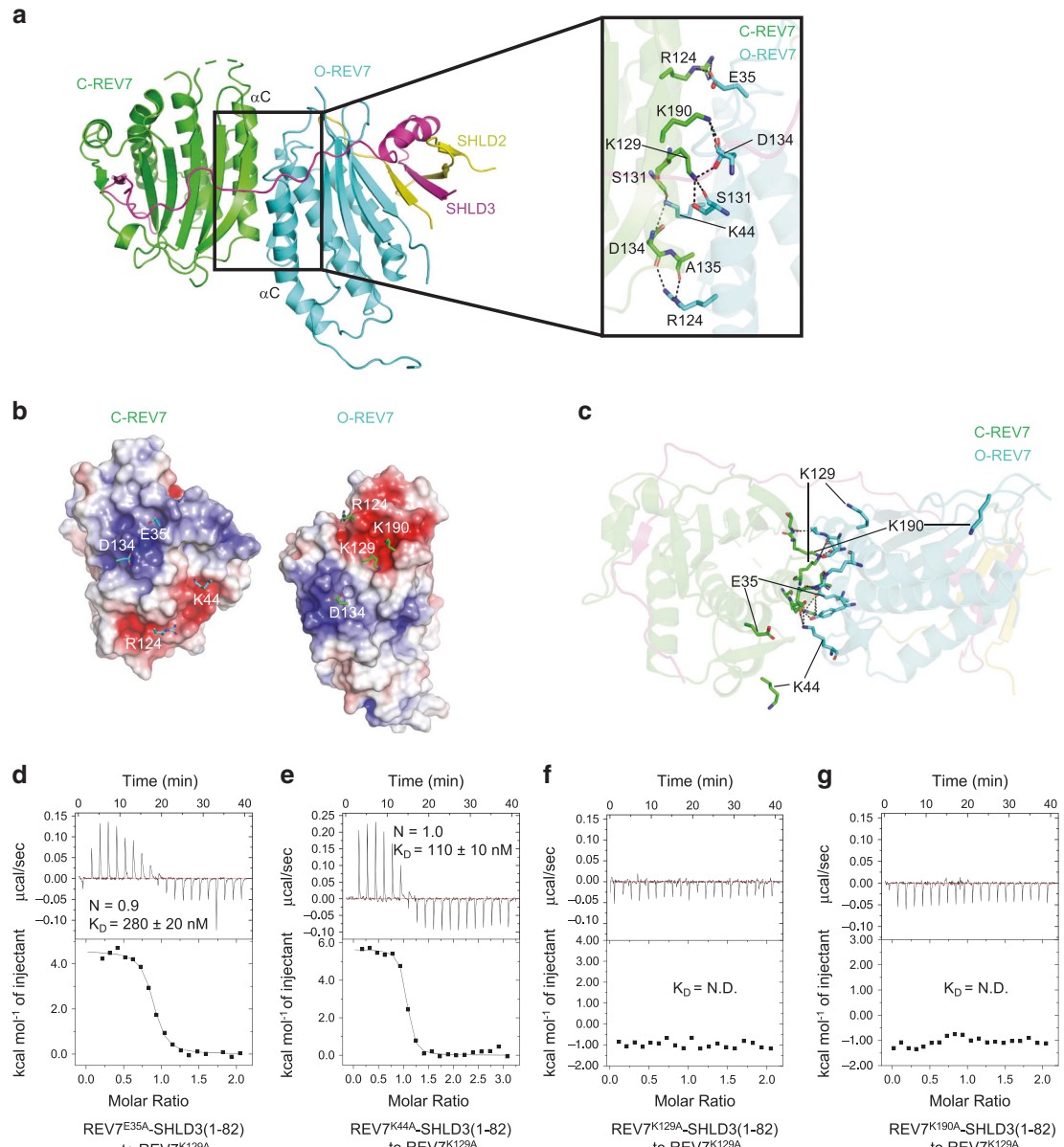

**Fig. 2 Structural basis of the asymmetric REV7 conformational dimer. a** Details of the hydrogen bond network in REV7 dimer interface. Hydrogen bonds are denoted as dashed black lines. **b** The electrostatic surface representation of the C-REV7-O-REV7 interface (positive potential, blue; negative potential, red). Key basic and acidic residues that interact with each other are shown in sticks. C-REV7 is viewed from O-REV7 along x axis and O-REV7 is viewed from C-REV7 along x axis based on Fig. 2a. In the left panel, C-REV7 is shown as electrostatic surface model and the amino acid residues of O-REV7 that interact with C-REV7 are shown in sticks. In the right panel, O-REV7 is shown in electrostatic surface model while residues of C-REV7 that interact with O-REV7 are shown in sticks. **c** Structural details of the dimer interface show the contribution of Glu35, Lys44, Lys129 and Lys190 is different between C-REV7 and O-REV7. **d**, **e** ITC measurements of binding affinities between REV7$^{E35A}$-SHLD3(1–82) or REV7$^{K44A}$-SHLD3(1–82) and REV7$^{K129A}$. The binding constants ($K_D$ values ± standard deviations) and stoichiometries (N) are indicated. $K_D$ value and standard deviation were calculated from three independent experiments. Source data are provided as a Source Data file. **f**, **g** ITC measurements of the interaction between REV7$^{K129A}$-SHLD3(1–82) or REV7$^{K190A}$-SHLD3(1–82) and REV7$^{K129A}$. REV7$^{K129A}$-SHLD3(1–82) and REV7$^{K190A}$-SHLD3(1–82) could hardly bind to REV7$^{K129A}$. N.D.: no detectable binding.

REV7$^{WT}$ binding capability, which leads to the deficiency of the shieldin complex assembly. Therefore, these results demonstrate that SHLD3 mediated REV7 conformational dimerization is essential for the recruitment of SHLD2 and efficient NHEJ.

**SHLD2 forms a β sheet sandwich with O-REV7 and SHLD3.** Our crystal structure demonstrates the structural basis of SHLD2 recognition by the SHLD3-O-REV7 complex. β1 of SHLD2 forms antiparallel β sheet with β5 of O-REV7 and parallel β sheet with

β1 of SHLD3. β2 of SHLD2 forms antiparallel β sheet with β1 of SHLD3 (Fig. 6a). Thus, β1 and β2 of SHLD2 form a sandwich with β1 of SHLD3, while β1 of SHLD3 and β5 of O-REV7 form a sandwich with β1 of SHLD2, which are mainly connected by hydrogen bonds between backbones of β strands (Fig. 6a, left panel). Furthermore, Arg10 of SHLD3 forms a salt bridge with Asp66 of O-REV7 and Tyr63 of O-REV7 forms a hydrogen bond with the backbone of SHLD2 (Fig. 6a, left panel). Despite this, Phe10 of SHLD2 makes hydrophobic interactions with Val71,

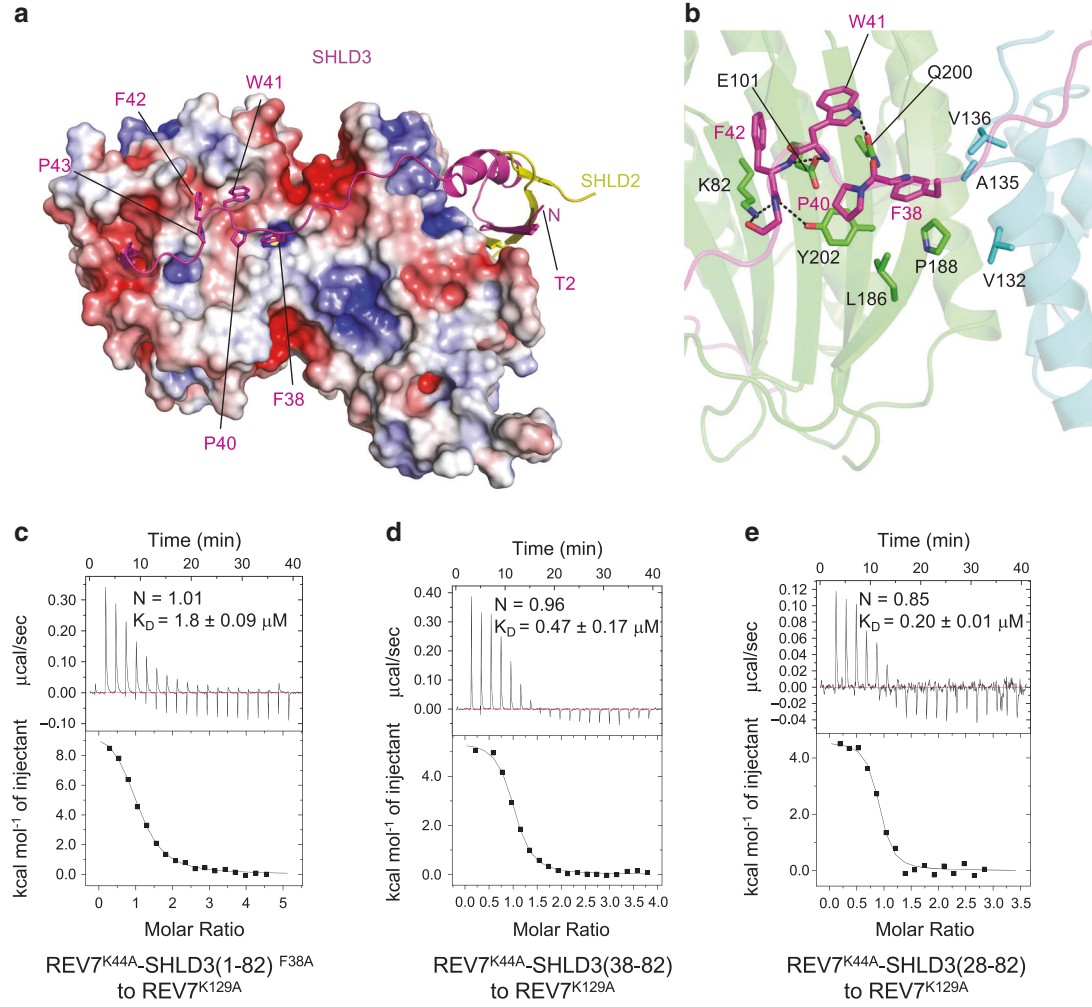

**Fig. 3 The highly conserved FXPWFP motif of SHLD3 binds to C-REV7 and enhances the binding affinity between C-REV7 and O-REV7. a** The interface between SHLD3 and C-REV7. C-REV7 and O-REV7 are shown in electrostatic surface representation (positive potential, blue; negative potential, red), SHLD3 in ribbon view, and the FXPWFP motif in sticks. The N terminus of SHLD3 is also indicated. Residues 1-37 of SHLD3 are shown in ribbon. **b** Close-up view of the interface between the FXPWFP motif (magenta) and C-REV7 (green), O-REV7 (cyan). Hydrogen bonds are indicated by dashed lines and the residues that are involved in the hydrophobic interactions are shown in sticks. **c–e** ITC measurements of interaction between distinct REV7[K44A]-SHLD3 (1–82) truncations or mutants and REV7[K129A]. The calculated N and $K_D$ are indicated as described in Fig. 2d. Source data are provided as a Source Data file.

Leu74, Val148 and Val150 of O-REV7 (Fig. 6a, right panel). On the reverse side, extensive hydrophobic interactions are made by lots of hydrophobic residues, including Val7, Ile9, Trp11, Ile39, Leu41, Tyr43, Leu48, Leu50 of SHLD2, Val5, Leu7, Tyr9, Pro16, Leu19, Pro20, Ile27 of SHLD3 and Leu149 of O-REV7 (Fig. 6b). The Cβ-Cδ of K37[SHLD2] is also involved in this interaction. Totally, the interface between SHLD2 and O-REV7 buries 1678 Å$^2$ area and an area of 2471 Å$^2$ is buried between SHLD2 and SHLD3 interface. Interestingly, the structural alignment also shows β1 of SHLD2 occupies the positon of RBM$_2$ of SHLD3 in C-REV7-SHLD3 and the N-terminal region of SHLD3 displaces the safety belt of C-REV7 (Fig. 6c). This unique binding mode and these extensive contacts make the complex very stable.

Furthermore, the amino acid residues between the parallel β sheet formed by β1 of SHLD2 and β1 of SHLD3 are similar (Supplementary Fig. 6a). Therefore, it is possible that without SHLD2, SHLD3 interacts with O-REV7 in a similar manner. To verify this, we performed molecular dynamics simulations by deletion of SHLD2. Notably, after simulation, SHLD3-C-REV7-O-REV7 forms a stable complex and β1 of SHLD3 binds to O-REV7 in a similar manner as β1 of SHLD2 (Fig. 6d and Supplementary Fig. 6b). Hence the N terminus (amino acids

1–27) of SHLD3 interacts with O-REV7 when it forms a conformational dimer with C-REV7. However, the binding affinity between SHLD3(1–27) and O-REV7 is extremely weak because deletion of SHLD3(1–27) has little impact on the interaction between REV7[K44A]-SHLD3 and REV7[K129A] as shown in Fig. 2e and Fig. 3e (0.11 ± 0.01 μM versus 0.20 ± 0.01 μM). Therefore, SHLD2 displaces β1 of SHLD3 to further lock the complex in a more stable state. These results show that SHLD2 acts as a bolt to lock SHLD3 and O-REV7 tightly.

**O-REV7-SHLD2 interaction impairment leads to NHEJ deficiency.** We further investigated how the interaction between O-REV7 and SHLD2 impact NHEJ efficiency. Tyr63 and Trp171 are two evolutionarily conserved residues that contribute equally to interaction between safety-belt and RBMs (Supplementary Fig. 7a). Our structure shows O-REV7 Tyr63 but not Trp171 locates at the interface between SHLD2 and O-REV7 (Fig. 7a). We suppose that REV7[Y63A] may influence the interaction between the SHLD3-REV7 conformational dimer and SHLD2. We purified the recombinant expressed REV7[Y63A]-SHLD3(1–82) and REV7[W171A]-SHLD3(1–82), which form homogeneous conformational dimer and both form stable complexes with

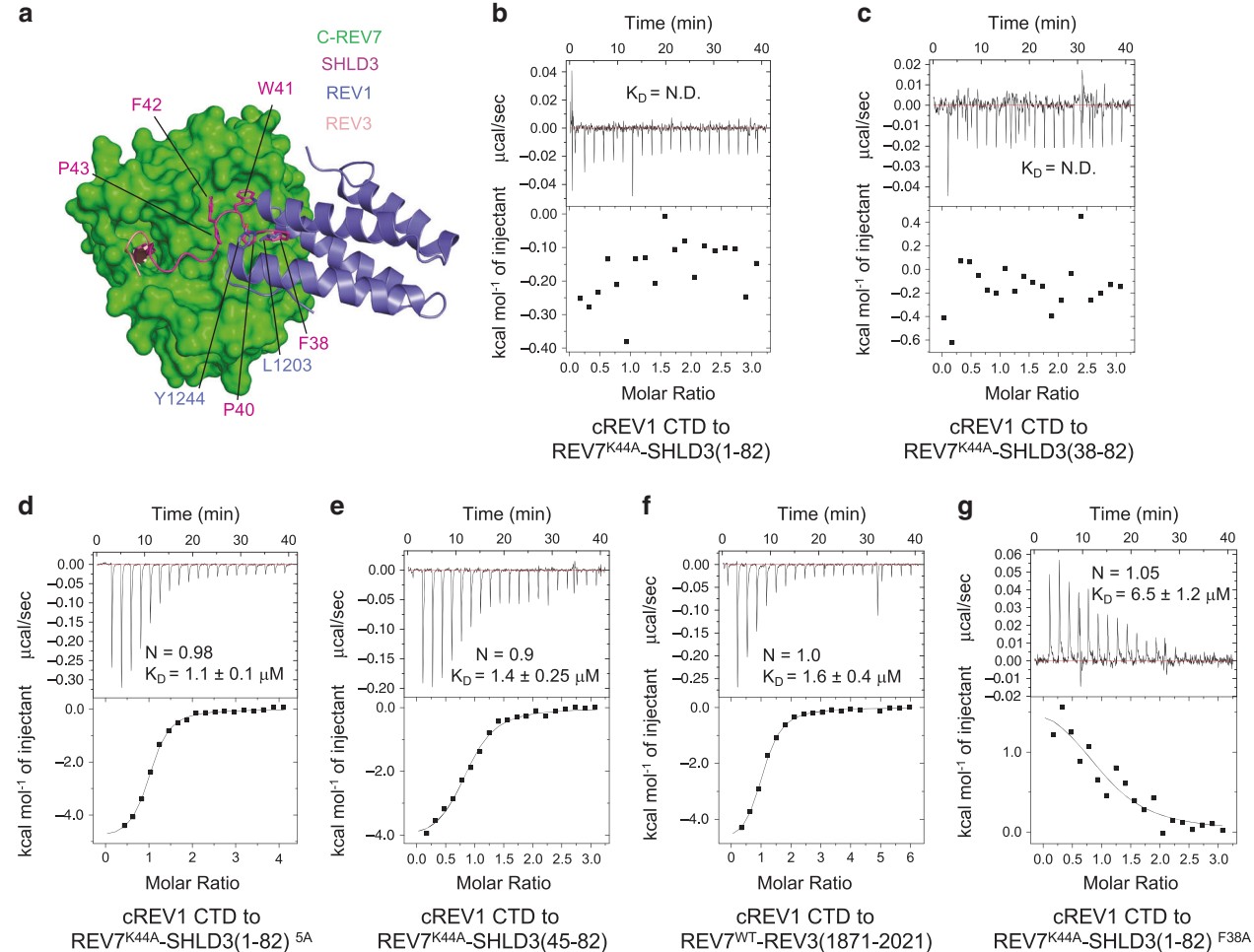

**Fig. 4 The FXPWFP motif masks the binding between REV1 and REV7. a** Structural superimposition of C-REV7 (green)–SHLD3 (magenta) and REV3-C-REV7-REV1 (PDB: 3VU7). The superimposition was done by align C-REV7. C-REV7 is shown in surface representation, REV1 and SHLD3 in ribbon view, and the FXPWFP motif in sticks. Residues Leu1203 and Tyr1244 of REV1 that contribute its interaction with REV7 are shown in sticks. **b–g** ITC measurements of interaction between cREV1 CTD and distinct REV7$^{K44A}$-SHLD3(1–82) truncations, mutants or REV7$^{WT}$-REV3(1871–2021). REV7$^{WT}$-REV3(1871–2021) is a representative C-REV7 that binds to cREV1 CTD. The calculated N and $K_D$ are indicated as described in Fig. 2d. N.D.: no detectable binding. Source data are provided as a Source Data file.

MBP-SHLD2(1–60) (Fig. 7b, Supplementary Fig. 7b, c and Supplementary Table 1). However, as shown in Fig. 7c, the binding affinity between REV7$^{Y63A}$-SHLD3(1–82) or REV7$^{W171A}$-SHLD3 (1–82) and MBP-SHLD2(1–60) is 600 ± 270 nM and 59 ± 20 nM, respectively. These results show that although REV7$^{Y63A}$-SHLD3 (1–82) still forms a stable complex with MBP-SHLD2(1–60), it has much weaker affinity when compared with REV7$^{W171A}$-SHLD3(1–82). Therefore, we propose that impaired ability of REV7$^{Y63A}$ to interact with SHLD2 may impair NHEJ efficiency. This is confirmed by our NHEJ assay that REV7$^{Y63A}$ significantly interferes with NHEJ while REV7$^{W171A}$ has moderate impact (Fig. 7d).

Then we tested the binding affinity of these two mutants with SHLD3(1–82) in vitro to exclude the influence of the SHLD3-REV7 interaction deficiency. As expected, SHLD3(1–82) binds to REV7$^{Y63A}$ and REV7$^{W171A}$ with an equivalent affinity (about 200 nM), which is just a little lower than REV7$^{WT}$ (about 130 nM) (Fig. 7e; Supplementary Fig. 7d). Afterwards, we also detected the interaction between SHLD3 and REV7 mutants in vivo, which even shows REV7$^{Y63A}$ exhibits stronger interaction with SHLD3 than REV7$^{W171A}$ (Fig. 7f, right panel, lane 3 versus lane 1). The interaction was enhanced after doxorubicin treatment which causes DSBs, possibly due to the upregulation

of SHLD3 (Fig. 7f, lane 2 versus lane 1, lane 4 versus lane 3). Meanwhile, the observation that REV7$^{W171A}$ shows weaker interaction with SHLD3 in vivo can be explained by its stronger interaction with REV3(1042–1251 + 1847–2021) (Fig. 7f, right panel, lane 3 versus lane 1). Furthermore, we also confirmed this in vitro. REV7$^{Y63A}$-SHLD3(1–82) shows much weaker affinity with MBP-REV3(1847–2021) than REV7$^{W171A}$-SHLD3(1–82) (37 ± 18 μM versus 1.1 ± 0.5 μM), and thus MBP-REV3 (1847–2021) cannot compete off SHLD3(1–82) in REV7$^{Y63A}$-SHLD3(1–82) as efficiently as SHLD3(1–82) in REV7$^{W171A}$-SHLD3(1–82) (Supplementary Fig. 7e, f). Taken together, through the analysis of REV7$^{Y63A}$ and REV7$^{W171A}$, impairment of the interaction between SHLD2 and O-REV7 disturbs NHEJ efficiency.

**REV3 interacts with SHLD3 mediated REV7 conformational dimer.** SHLD3 mediated REV7 conformational dimer makes a completely open REV7. Given REV3 interacts with O-REV7 to form Pol ζ[12], we wondered if the O-REV7 in the conformational dimer could interact with REV3. We measured the binding affinity of REV3(1847-1906) with the reconstituted REV7$^{E35A}$-SHLD3(1–82)-REV7$^{K129A}$ conformational dimer, which shows a strong interaction with the equilibrium dissociation constant to

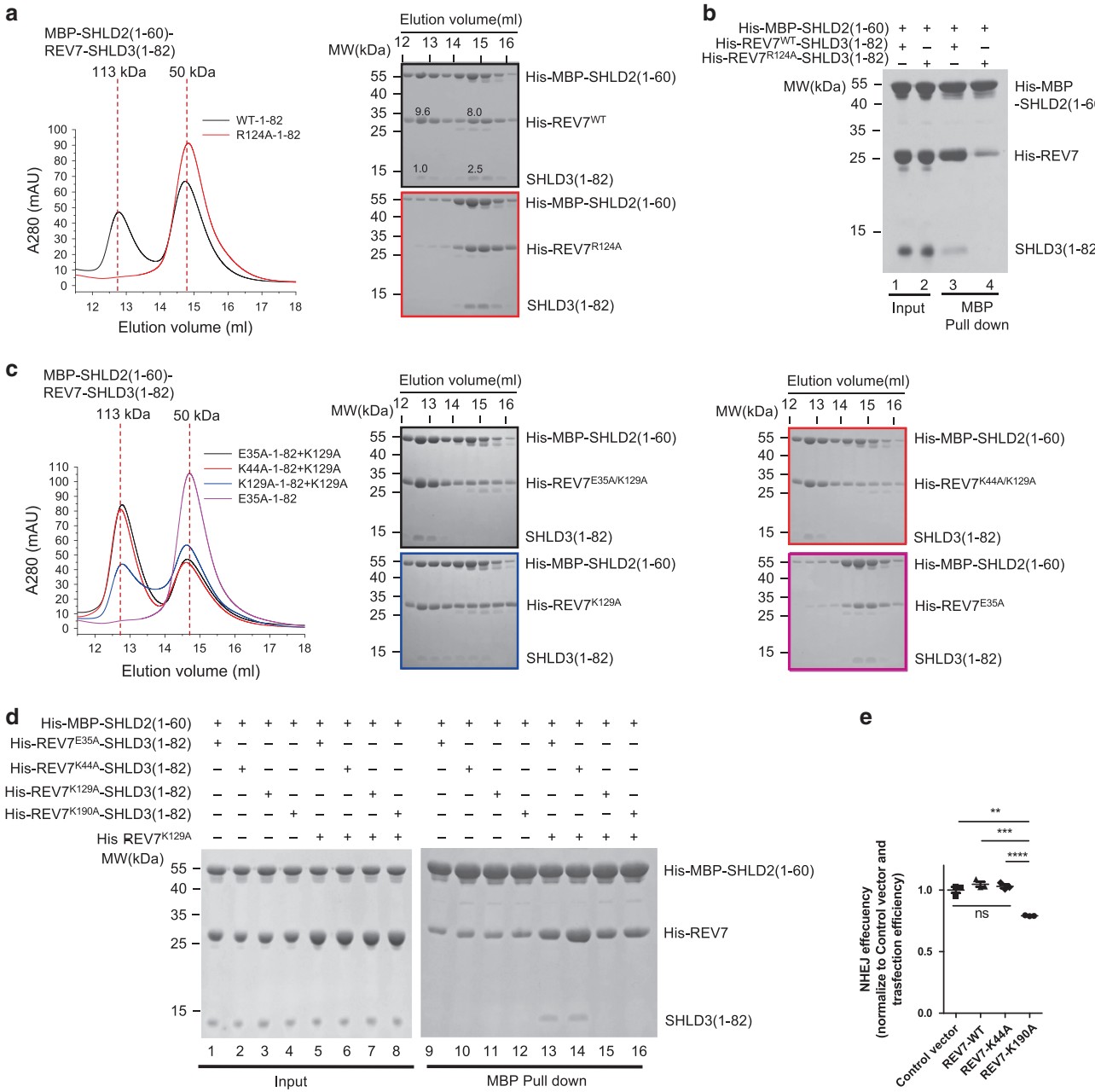

**Fig. 5 SHLD3 mediated REV7 conformational dimer is essential for the recruitment of SHLD2. a** Gel filtration profiles show the interaction between REV7[WT/R124A]-SHLD3(1–82) (short as WT/R124A-1–82 in the Figure) and MBP-SHLD2(1–60) in a Superdex200 Increase 10/300 size exclusion chromatography (SEC) column (the left panel). MBP-SHLD2(1–60) is excessive. The peaks eluted at 12.5–13 ml are composed of the stable complex of MBP-SHLD2(1–60)-REV7-SHLD3(1–82) while the peaks eluted at 14.5–15 ml are un-complexed MBP-SHLD2(1–60) or REV7-SHLD3(1–82). Fractions (0.5 ml each) corresponding to REV7[WT]-SHLD3(1–82) and MBP-SHLD2(1–60) co-elution (black), REV7[R124A]-SHLD3(1–82) and MBP-SHLD2(1–60) co-elution (red) were analyzed by SDS-PAGE and stained by Coomassie brilliant blue (the right panel). ($n = 2$). **b** MBP pulldown showing the interaction between MBP-SHLD2(1–60) and REV7[WT/R124A]-SHLD3(1–82). Samples were analyzed by SDS-PAGE and stained by Coomassie brilliant blue. ($n = 4$). **c** Gel filtration profiles show the interaction between REV7 mutants and MBP-SHLD2(1–60). Various mutants of REV7-SHLD3(1–82) complex (200 μg) were first incubated with or without REV7[K129A] (short as K129A in the Figure) (200 μg), after 10 min, excessive MBP-SHLD2(1–60) was added. Fractions (0.5 ml each) corresponding to REV7[E35A]-SHLD3(1–82)-REV7[K129A] (short as E35A-1–82 + K129A in the Figure) and MBP-SHLD2(1–60) co-elution (black), REV7[K44A]-SHLD3(1–82)-REV7[K129A] (short as K44A-1–82 + K129A in the Figure) and MBP-SHLD2(1–60) co-elution (red), REV7[K129A]-SHLD3 (1–82)-REV7[K129A] (short as K129A-1–82 + K129A in the Figure) and MBP-SHLD2(1–60) co-elution (blue), REV7[E35A]-SHLD3(1–82) (short as E35A-1–82 in the Figure) and MBP-SHLD2(1–60) co-elution (pink) were analyzed by SDS-PAGE and stained by Coomassie brilliant blue (the right panel). ($n = 2$). **d** MBP pulldown assay shows the interaction between MBP-SHLD2(1–60) and REV7-SHLD3(1–82) mutants or reconstituted conformational dimer. Samples were analyzed by SDS-PAGE and stained by Coomassie brilliant blue. ($n = 2$). **e** NHEJ efficiency was determined in Hela cells overexpressing exogenous FLAG-tagged wild-type REV7, REV7-K44A, REV7-K190A or control vector. Data were analyzed with the unpaired two-tailed Student's test. Error bars indicate SEM ($n = 3$). Ns, no significant difference, **$p < 0.01$, ***$p < 0.001$, ****$p < 0.0001$. Exact p-values are 0.0011 (Control versus REV7-K190A), 0.00024 (REV7-WT versus REV7-K190A), 0.000093 (REV7-K44A versus REV7-K190A), 0.22 (Control versus REV7-WT), 0.36 (Control versus REV7-K44A) and 0.54 (REV7-WT versus REV7-K44A). n, biologically independent experiments. Source data are provided as a Source Data file.

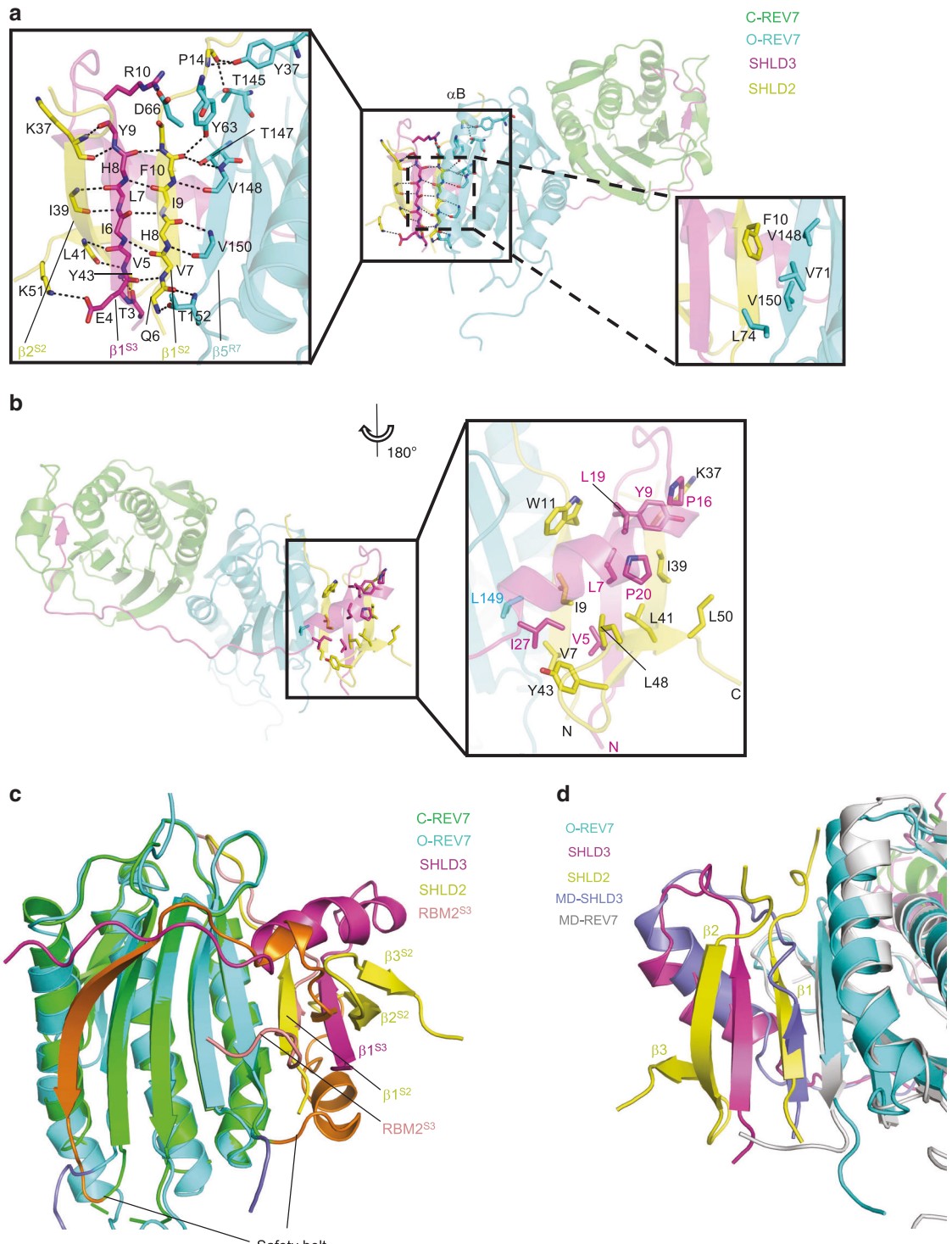

**Fig. 6 Structural mechanism of SHLD2 recognition by O-REV7-SHLD3. a** Details of the hydrogen bond network (the left panel) and hydrophobic network (the right panel). In the detailed view, αB of O-REV7 and side chains that do not contribute to polar contacts are omitted for convenience to read. **b** Hydrophobic network in the reverse side rotated along y-axis compared with Fig. 6a. **c** Structural alignment of C-REV7-RBM$_2$$^{S3}$ and O-REV7-SHLD2-SHLD3. The regions between two lines represent safety belt and are colored in orange (C-REV7) and slate (O-REV7). RBM$_2$$^{S3}$ represents RBM$_2$ of SHLD3 and is colored in salmon. The superscript S2 and S3 represents SHLD2 and SHLD3, respectively. **d** Molecular dynamics simulation results of the SHLD3-C-REV7-O-REV7 complex and structural alignment of SHLD3-REV7-SHLD2 complex and SHLD3-C-REV7-O-REV7 complex shows the interaction model of SHLD3 β1 and O-REV7. MD-SHLD3 and MD-REV7 shows structures of SHLD3 and O-REV7 after molecular dynamics simulation. The secondary structures of SHLD2 are labeled in detail.

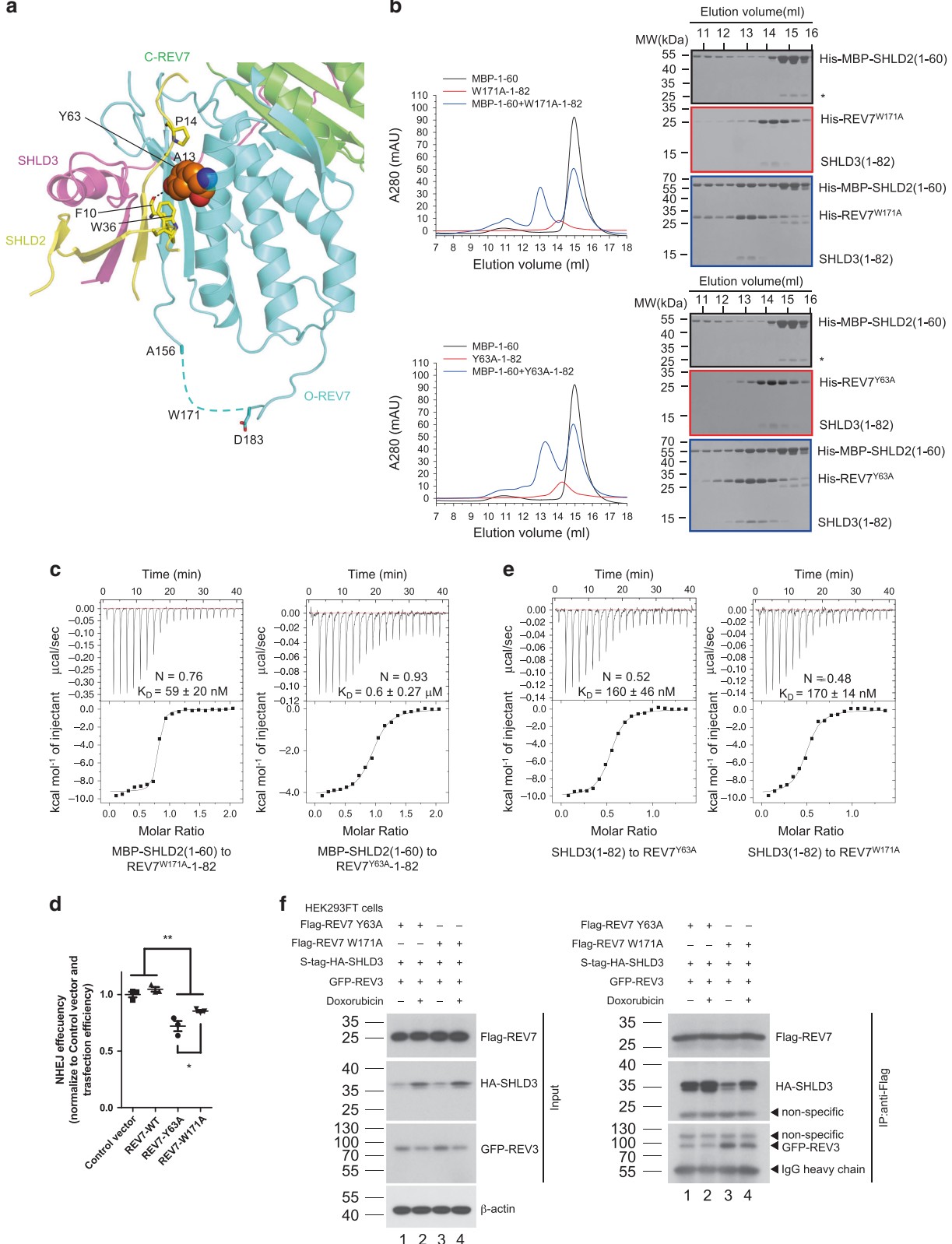

be $46 \pm 10$ nM (Fig. 8a). After the titration, we injected samples into a Resource Q column. As shown in Supplementary Fig. 8a, the conformational dimer is dissociated and forms two new complexes in the context of ion exchange solution, one is composed of REV7 and SHLD3(1–82), while the other contains REV7 and REV3(1847-1906). To verify that REV3(1847-1906) indeed binds to O-REV7, we distinguished two states of REV7 by

reconstituting the conformational dimer with MBP-REV7$^{K129A}$ (O-REV7) and REV7$^{E35A}$-SHLD3(1–82). Expectedly, MBP-REV3 (1847-1906) elutes with MBP-REV7$^{K129A}$ together, while SHLD3 (1–82) co-elutes with REV7$^{E35A}$, which indicates that upon binding to REV3, O-REV7 is dissociated from the conformational dimer and forms a C-REV7-REV3 dimer (Fig. 8b). This is in accordance with the much lower binding affinity between

**Fig. 7 REV7$^{Y63A}$ shows impaired ability to interact with SHLD2 but not SHLD3. a** Structural details of the interface between O-REV7 and SHLD2. C-REV7 is shown in green, O-REV7 is shown in cyan, SHLD3 is shown in magenta and SHLD2 is shown in yellow. Tyr63 of O-REV7 are shown in spheres model to highlight its interaction with SHLD2. The residues Phe10, Ala13, Pro14 and Trp36 of SHLD2 that make interactions with Tyr63 are shown in sticks. Trp171 of O-REV7 locates in the disordered region shown as dashed lines. **b** Gel filtration profiles show the interaction between MBP-SHLD2(1–60) (short as MBP-1–60) and REV7$^{Y63A}$-SHLD3(1–82)/REV7$^{W171A}$-SHLD3(1–82) (short as Y63A-1–82/W171A-1–82) in a Superdex200 Increase 10/300 SEC column. MBP-SHLD2(1–60) is excessive. Abs 0.1% of REV7$^{Y63A}$-SHLD3(1–82), REV7$^{W171A}$-SHLD3(1–82), and MBP-SHLD2(1–60) are 0.534, 0.405, and 1.668, respectively. This indicates extinction coefficient of REV7-SHLD3(1–82) is low, which accounts for its lower A280. Asterisk represents impurity. ($n = 2$). **c** ITC measurements of interaction between REV7$^{Y63A}$-SHLD3(1–82) or REV7$^{W171A}$-SHLD3(1–82) and MBP-SHLD2(1–60). The calculated N and $K_D$ are indicated as described in Fig. 2d. **d** NHEJ efficiency was determined in Hela cells overexpressing exogenous FLAG-tagged wild-type REV7, REV7-Y63A, REV7-W171A or control vector. Data were analyzed with the unpaired two-tailed Student's test. Error bars indicate SEM ($n = 3$). $*p < 0.05$, $**p < 0.01$. Exact $p$-values are 0.0056 (Control versus REV7-Y63A), 0.0054 (Control versus REV7-W171A), 0.0028 (REV7-WT versus REV7-Y63A), 0.0011 (REV7-WT versus REV7-W171A) and 0.045 (REV7-Y63A versus REV7-W171A). **e** ITC measurements of interaction between SHLD3(1–82) and REV7$^{Y63A}$ or REV7$^{W171A}$. The calculated N and $K_D$ are indicated as described in Fig. 2d. **f** S-tag-HA-SHLD3 and Flag-REV7 mutants were co-expressed with GFP-tagged REV3(1042–1251 + 1847–2021) (short as GFP-REV3 in the Figure) in HEK293FT cells. Indicated cells were treated with DMSO, or 1 μM doxorubicin for 24 h, then lysed with NP-40 lysis buffer. Flag-REV7 and its associated proteins were purified by anti-Flag agaroses. HA, Flag and GFP antibodies were used to detect S-tag-HA-SHLD3, Flag-REV7 and GFP-tagged REV3(1042–1251 + 1847–2021), respectively. The input panel shows the transfection efficiency and the IP panel shows the interaction between REV7 mutants and SHLD3 or REV3(1042–1251 + 1847–2021). ($n = 2$). $n$, biologically independent experiments. Source data are provided as a Source Data file.

REV7$^{E35A}$-SHLD3(1–82) and REV7$^{K129A}$-REV3(1847-1906) (represents C-REV7) ($K_D = 6.6 \pm 2.4$ μM) as compared with REV7$^{E35A}$-SHLD3(1–82) and REV7$^{K129A}$ ($K_D = 0.28 \pm 0.02$ μM). (Fig. 8c and Fig. 2d). Since REV3(1847–2021) has two binding sites for REV7 with 10 nM level binding affinity, we further tested the binding ability between REV7-SHLD3(1–82) and MBP-REV3 (1847–2021). As shown in Fig. 8d, MBP-REV3(1847–2021) efficiently binds O-REV7 in the REV7-SHLD3(1–82) complex while could not compete with SHLD3 for the C-REV7. This is because both SHLD3 and REV3 binds to REV7 through the PXXXpP motif with a similar binding affinity to form C-REV7 (Supplementary Fig. 8b, c).

Afterwards, we also detected the interaction between SHLD3 and REV3(1847–2021) in vivo and the interaction is enhanced after doxorubicin treatment that causes DSBs (Fig. 8e, lane 4 versus lane 2). Meanwhile, to determine whether REV3 accumulates at DSB sites in living cells, we induced DNA damage tracts (laser lines) by employing laser micro-irradiation. REV3$^{TR1}$ is an approximately minimal truncation of REV3 that remains its normal function according to previous report[28] (Fig. 8f). Both full length and the functional truncated GFP-tagged REV3 (REV3$^{TR1}$) accumulated in laser-lines, which were discerned by their co-localization with mCherry-tagged SHLD3 (Fig. 8g, h). These results suggest that REV3 accumulates at DNA damage sites and SHLD3 mediated REV7 conformational dimer interact with REV3 both in vitro and in vivo and that SHLD3 mediated REV7 conformational dimer may act as a platform to coordinate various proteins to accomplish NHEJ.

## Discussion

Shieldin is an important downstream effector of 53BP1-RIF1 to regulate the repair of DNA DSBs. In this study, we successfully solved the crystal structure of the SHLD3-REV7-SHLD2 complex, which reveals a striking SHLD3 mediated conformational dimer of C-REV7-O-REV7 and established the molecular architecture of the shieldin complex (Supplementary Fig. 8d).

The previous report shows that the interaction between SHLD3 and REV7 is abolished by REV7$^{Y63A}$ but unaffected by REV7$^{W171A}$ in yeast two-hybrid (Y2H) assay[6]. However, our structural and biochemical studies demonstrate that Tyr63 and Trp171 of REV7 play equal important roles in the REV7-SHLD3 interaction. This controversial situation may be a result of the technical limitation of the Y2H assay. Our structural analysis show that SHLD3 makes much more extensive contacts with REV7 than the RBM$_1$ of REV3 does with REV7. Thus, neither REV7$^{Y63A}$ nor REV7$^{W171A}$ mutant abolishes the interaction

between REV7 and SHLD3. On the other hand, O-REV7 Tyr63 but not Trp171 locates at the interface between SHLD2 and O-REV7, and REV7$^{Y63A}$ diminishes its interaction with SHLD2 while REV7$^{W171A}$ has no effect. This is consistent with the phenomenon previously reported that cells expressing REV7$^{Y63A}$ exhibit lower CSR efficiency than those expressing REV7$^{W171A}$ mutant[6]. Furthermore, it is reported that REV7$^{K129A}$ fails to interact with SHLD1-SHLD2 in vivo and shows CSR deficiency[6]. Our studies show that Lys129 of REV7 is essential for the integration of the REV7 conformational dimer and this conformational dimer is essential for the interaction with SHLD2 can perfectly explain the CSR deficiency in cells expressing REV7$^{K129A}$ mutant. Moreover, we found that SHLD3(1–27) is also essential for priming SHLD2 binding to O-REV7.

Our results show that SHLD3 mediated REV7 conformational dimer not only recruits SHLD2, but also interacts with other proteins like REV3 that can bind to O-REV7. This suggests that the SHLD3-C-REV7-O-REV7 trimer acts as a platform to recruit various proteins other than only SHLD2 for exerting different cellular functions, which is in accordance with the observation that SHLD3 and REV7 have similar abundance while SHLD1 and SHLD2 have a much lower abundance in the affinity-purification mass spectrometry data[1]. Since both Mad2 and REV7 form a conformational dimer to recruit downstream effectors, this may be a universal mechanism utilized by HORMA proteins[26]. It is known that Mad2 is regulated by p31$^{comet}$ and ATPase TRIP13 through structural remodeling[29,30]. However, how shieldin is regulated is unknown. Whether shieldin is regulated in a similar manner needs to be tested. Since disruption of the conformational dimer disables the assembly of the SHLD3-REV7-SHLD2 complex and TRIP13 is found to interact with REV7[4], it is possible that TRIP13 regulates shieldin through structural remodeling of REV7 to disrupt the conformational dimer. Meanwhile, since mutants that abolish the conformational dimer (such as R124A, K129A) show stronger interaction with SHLD3, these mutants could function in dominant-negative manner.

CTC1–STN1–TEN1 (CST) –Pol α has been found to be recruited to DSBs in a 53BP1- and shieldin-dependent manner to mediate fill-in reaction[5]. However, Pol α-primase can only synthesize about 20 nt[31], but the ssDNA tails in DSBs are usually longer than 20 nt[32]. Thus, the poor processivity of Pol α-primase seems to be problematic to ensure the complement of ssDNA during the process of NHEJ, suggesting that more processive polymerases may be involved after synthesis is initiated. Pol ζ is more processive than Pol α[28], ablation of which in B cells impairs class switch recombination and DNA break repair that is verified

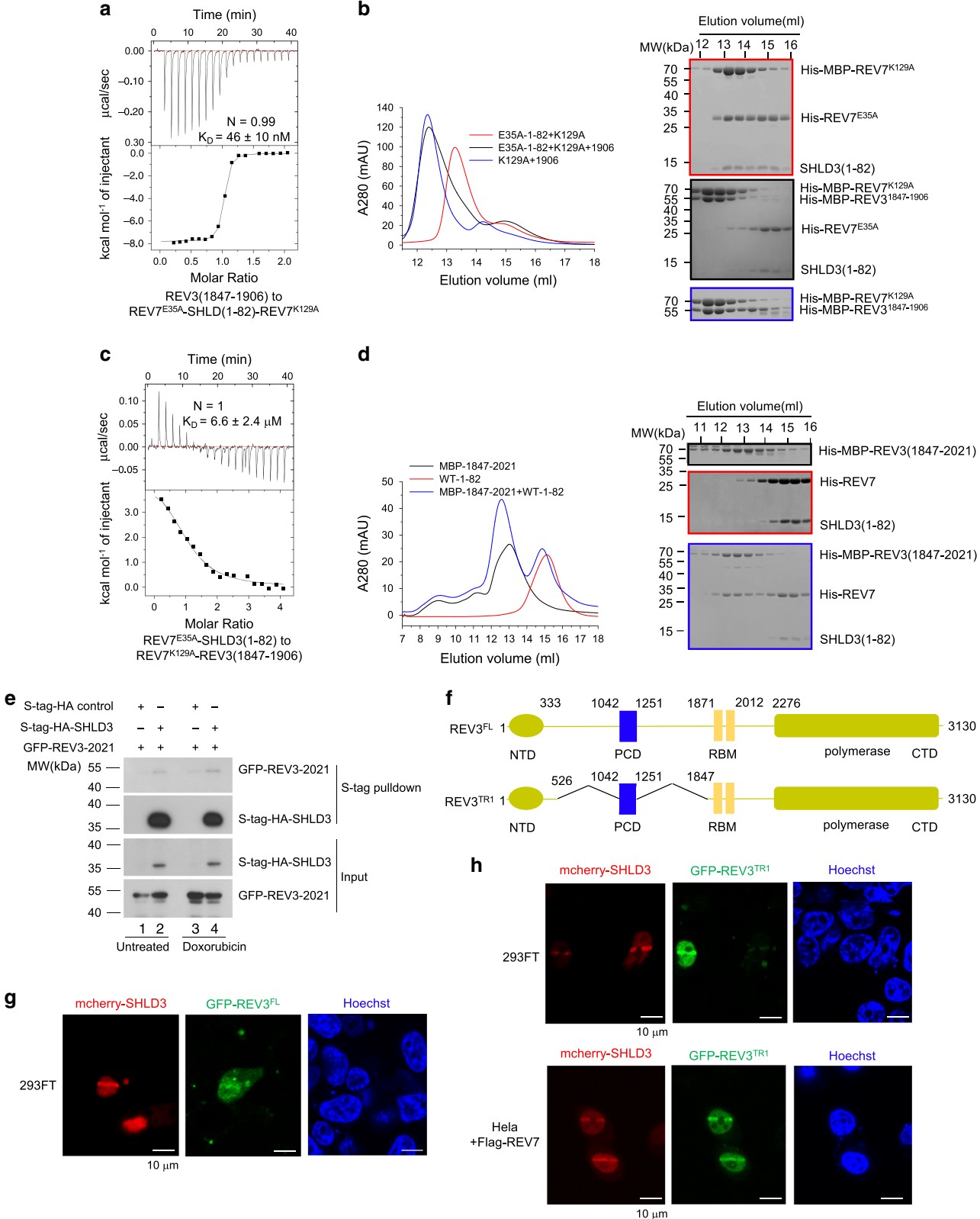

by conditional knockout of *Rev3*[10,33]. CH12 cells with *Rev3* depletion exhibits normal RPA recruitment but CSR deficiency[10], indicating that *Rev3* ablation does not affect the protection of ssDNA, and suggesting that REV3 is essential for the mediated step(s) between ssDNA protection and the final step of NHEJ. Given the fact that Pol ζ has been designated as an extender polymerase, and our results show its catalytic subunit REV3 can

be recruited to DSB sites, we propose that Pol ζ may orchestrate with Pol α to complement ssDNA to double-strand DNA, especially when the ssDNA is longer than 20 nt, which is beyond the processive capacity of Pol α, to trigger NHEJ.

Pol ζ/REV1 plays an essential role in the TLS that usually leads to resistance of cancer cells to chemotherapy[34]. REV1 CTD interact with Pol ζ through the REV7 subunit and inhibition of

**Fig. 8 REV3 interacts with SHLD3 mediated C-REV7-O-REV7 and locates at DSB sites. a** ITC measurement of interaction between REV7[E35A]-SHLD3 (1–82)-REV7[K129A] and REV3(1847-1906). REV7[K129A] was first saturated by excessive REV7[E35A]-SHLD3(1–82) as shown in Fig. 2d. The calculated N and $K_D$ are indicated as described in Fig. 2d. **b** Gel filtration profiles show the interaction between MBP-REV3(1847-1906) and REV7[E35A]-SHLD3(1–82)-MBP-REV7[K129A] (short as 1906, E35A-1–82 and K129A respectively in the figure). MBP-REV7[K129A] was first saturated by excessive REV7[E35A]-SHLD3(1–82) as shown in red line. Co-elutions were analyzed by SDS-PAGE and stained by Coomassie brilliant blue. **c** ITC measurement of interaction between REV7[E35A]-SHLD3(1–82) and REV7[K129A]-REV3(1847-1906). REV7[K129A] was first saturated by excessive REV3(1847-1906). The calculated N and $K_D$ are indicated as described in Fig. 2d. **d** Gel filtration profiles show the interaction between MBP-REV3(1847–2021) and REV7[WT]-SHLD3(1–82) (short as MBP-1847–2021 and WT-1–82 in the figure). Co-elutions were analyzed by SDS-PAGE and stained by Coomassie brilliant blue. **e** S-tag-HA-SHLD3 was co-expressed with GFP-tagged REV3(1847–2021) (short as GFP-REV3-2021 in the figure) in HEK293T cells. Indicated cells were treated with DMSO or 1 μM doxorubicin for 24 h. S-tag-HA-SHLD3 and its associated proteins were purified by S-protein beads. HA and GFP antibodies were used to detect S-tag-HA-SHLD3 and GFP-tagged REV3(1847–2021), respectively. The input panel shows the transfection efficiency and the S-tag pulldown panel shows the interaction between SHLD3 and REV3(1847–2021). Source data are provided as a Source Data file. **f** Schematic representation showing conserved domains of human REV3 and the truncation used in laser micro-irradiation assay. The N-terminal domain (NTD) and polymerase domain are shaded in yellow green, REV7 binding motif (RBM) in yellow, and the positively charged domain (PCD) in blue. Domain boundaries are indicated by residue numbers. In the REV3 deletion construct REV3[TR1], predicted unstructured portions with boundaries marked by residue numbers were deleted. FL: full length. **g**, **h** mCherry-SHLD3 and GFP-REV3 localization to DNA damage were monitored after laser micro-irradiation of 293FT and HeLa cells. Cells were imaged 5 min after damage induction except REV3[FL] (10 min) due to its large size. n = 2 biologically independent experiments in **b**, **d**, **e** and **h**, except **g** was assessed once.

the REV1 CTD-REV7 interaction is an attractive and possible avenue to improve chemotherapy[27]. Recently, a small molecule, JH-RE-06, was discovered to block the REV1-REV7 interaction by inducing REV1 dimerization[27]. JH-RE-06 shows promising effects to improve chemotherapy both in vitro and in vivo, indicating that inhibition of the REV1 CTD-REV7 interaction has therapeutic potential. Our structure shows the highly conserved FXPWFP motif blocks the REV7-REV1 CTD interaction by masking the REV1 CTD-binding surface of REV7, here we propose this unrevealed REV7 surface will help develop inhibitors to improve chemotherapy, and high-affinity inhibitors targeting this unrevealed REV7 surface is worth further development.

Taken together, our results reveal the unexpected architecture of SHLD3-C-REV7-O-REV7-SHLD2 tetramer, which provides new biological insight into how SHLD3-REV7 coordinates with different downstream mediators and how to develop new small molecules for improving chemotherapy.

## Methods

**Antibodies and chemicals.** The following antibodies were used for immuno-blotting: anti-Flag (Sigma-Aldrich, F3165), anti-HA (Sigma-Aldrich, H3663) and anti-GFP (Ray Antibody, RM1008). Doxorubicin (also named as Adriamycin, Selleck Chemicals, S1208), Anti-Flag M2 Affinity Gel (Sigma-Aldrich, A2220) and Hoechst 33342 (Biodee, DE0759) were purchased from the indicated sources. Plasmids are listed in Supplementary Table 2 and Supplementary Table 3.

**Cloning.** cDNAs encoding human REV7 and SHLD2 were kind gifts from Han lab. cDNA of REV7 full length was cloned into ORF1 of pETduet by EcoRI and NotI. For co-expression with REV7 and SHLD3, the coding sequences of truncations of SHLD2 were cloned into pET28a(+) by NcoI and XhoI without tag. For biochemical studies, the coding sequences of SHLD2(1–60), REV3(1847-1906) and REV3(1847–2021) were cloned into a modified pET28a(+) with His$_6$-MBP tag. For co-expression with REV7, the coding sequence of SHLD3 was synthesized by Synbio Technologies and truncations of it were cloned into ORF2 of REV7 inserted pETduet by NdeI and XhoI without His-tag. For in vivo studies, the coding sequence of SHLD3 was cloned into pmCherryC1 with an N-terminal mCherry tag or pCDH-puro with an N-terminal S-tag-HA tag. For biochemical studies, the coding sequences of SHLD3(1–82) and cREV1 CTD were cloned into pET28a(+) with a C-terminal His$_6$-tag. The coding sequence of the truncated REV3 was amplified from the cDNA of Jurkat cells and truncations of it was cloned into pET28a(+) with a C-terminal His$_6$-tag or pEGFPC1 with an N-terminal GFP tag. The coding sequence of full length REV3 was amplified from JT113-pETDuet1-(R)-hREV3L, which was a gift from Richard Wood (Addgene plasmid # 64872)[35]. Mutants of REV7 and SHLD3 were generated by site-directed mutagenesis. All plasmids were verified by sequencing (Ruibiotech).

**Expression and purification.** Expression vectors were transformed into *BL21* (DE3). For expressing the REV7- SHLD3(1–64)-SHLD2(1–52) complex, pET28a (+)-SHLD2(1–52) was co-transformed into *BL21*(DE3) with pETduet-REV7-SHLD3(1–64). Protein expression was induced with 0.1 mM isopropyl-1-thio- β-d-galactopyranoside (IPTG) at 20 °C. Cells expressed proteins were lysed in 20 mM

Tris, pH 7.4, 140 mM NaCl, 5 mM KCl, 10% glycerol, 1 mM Tris(2-carboxyethyl) phosphine (TCEP) and 1 mM phenylmethanesulfonylfluoride (PMSF) using ultrasonic cell crusher (XinChen) and centrifugation at 20,000 r.p.m. for 30 min. The supernatant was applied to a Ni-IDA beads (Smart-Lifesciences) and washed with buffer containing 20 mM Tris, pH 8.0, 500 mM NaCl, 1% glycerol and 0.5 mM TCEP with appropriate concentrations of imidazole. After that, proteins were eluted with elution buffer containing 20 mM Tris, pH 8.0, 500 mM NaCl, 1% glycerol, 0.5 mM TCEP and 300 mM imidazole. Proteins were concentrated to 500 μl and loaded onto a Superdex200 Increase 10/300 (GE Healthcare) equilibrated with gel filtration buffer containing 20 mM Tris, pH 8.0, 150 mM NaCl, 1% glycerol and 1 mM TCEP. Peaks containing proteins were collected and concentrated to a small volume. Protein concentrations were determined with microspectrophotometry using the theoretical molar extinction coefficients at 280 nm, and protein purity was evaluated with Coomassie blue staining of SDS–PAGE gels. Samples were flash frozen with liquid nitrogen in aliquots of 40 μl and stored at −80 °C until use. All proteins used in this study were purified using the same methods and buffer as described above.

**Crystallization and data collection.** The ternary complex crystals were obtained by sitting drop vapor diffusion in 0.02 M magnesium chloride hexahydrate, 0.1 M HEPES 7.5, 22% w/v Poly (acrylic acid sodium salt) 5100 at 20 °C with a concentration of 5 mg/ml. For data collection, the crystals were rapidly dipped in reservoir solution with 25% ethylene glycol and were flash frozen with liquid nitrogen. X-ray diffraction data were collected at beamline BL18U1 at Shanghai Synchrotron Radiation Facility (SSRF). The diffraction data were processed using HKL2000 and the CCP4 program suite[36].

**Structure determination.** The structure was determined by molecular replacement with Phaser[37] using the known REV7 structure (PDB ID 3VU7)[23] as the starting model. Multiple rounds of manual building and refinement were then performed using COOT[38] and PHENIX[39]. Densities of SHLD3 and SHLD2 become more and more clear during the refinement and the model of them were manually built. Diffraction data, refinement statistics, and quality of the structure are summarized in Table 1. The areas of the interfaces were calculated using the PISA server[40].

**Size-exclusion chromatography multi-angle light scattering.** SEC–MALS analysis was performed on a high-pressure injection system (Wyatt Technology) and chromatography system equipped with a DAWN HELEOS-II MALS detector and an Optilab T-rEX differential refractive index detector. An aliquot of 100 μl protein at 5 mg/ml was loaded onto a WTC-015S5 column (7.8 × 300 mm, 5 μm, Wyatt Technology) and eluted in buffer (20 mM Tris-HCl, pH 8.0, 150 mM NaCl, 1% glycerol, 1 mM TCEP, 0.01% NaN$_3$) at a flow rate of 0.4 ml/min. The outputs were analyzed by the ASTRA VI software (Wyatt Technology). The molecular mass was determined using the Astra 6 software program (Wyatt Technology) from the Raleigh ratio calculated by measuring the static light scattering and corresponding protein concentration of a selected peak.

**Isothermal titration calorimetry (ITC) measurements.** The interactions among the complexes were thermodynamically characterized using isothermal titration calorimetry on an ITC200 instrument (Malvern Instruments). All measurements were done in ITC buffer containing 20 mM Tris pH 8.0, 150 mM NaCl, 1% glycerol and 1 mM TCEP at 25 °C. Each titration consisted of 20 successive injections (the first at 0.4 μl and the remaining 19 at 2 μl). The heating power per injection was recorded and plotted as a function of time. The background was deduced either by the last several saturated titrations or by ligand-to-buffer titration (For those

titrations that could not be saturated). The binding isotherms were fitted to a one set of sites model using the MicroCal software. The stoichiometry of binding ($N$) and the equilibrium-association constant ($K_A$) were obtained directly. The equilibrium-dissociation constant ($K_D$) were derived from $K_A$.

**Size-exclusion chromatography**. Size-exclusion chromatography runs were performed on a Superdex 200 Increase 10/300 GL column (GE healthcare) using buffer (20 mM Tris/HCl pH 8.0, 150 mM NaCl, 1% glycerol and 1 mM TCEP). 400 μg of REV7$^{WT}$-SHLD3(1–82), REV7$^{R124A}$-SHLD3(1–82) or other mutant heterodimers were mixed with 400 μg of MBP-SHLD2(1–60) and incubated for 10 min on ice prior to the SEC run. For the reconstitution of the conformational dimer, 200 μg of REV7-SHLD3(1–82) mutants were first mixed with 200 μg REV7$^{K129A}$ to pre-form the complex, and then 400 μg of MBP-SHLD2(1–60) was added prior to the SEC run. The other runs were done by mixing 400 μg of two proteins prior to the SEC run. Samples were all loaded on the same column with a volume of 100 μl. The fractions (at a volume of 0.5 ml) obtained from the SEC runs were analyzed by 13% SDS-PAGE and stained by Coomassie brilliant blue.

**Anion exchange chromatography**. Prior to the anion exchange chromatography run, the concentration of NaCl was diluted to 80 mM. Buffer A was 20 mM Tris pH 8.0, 20 mM NaCl, 2 mM DTT, buffer B was 20 mM Tris pH 8.0, 500 mM NaCl, 2 mM DTT. The sample was eluted with the following gradient: 0–10% 2 ml, 10–50% 30 ml, 50–100% 10 ml. The fractions (at a volume of 1.0 ml) were analyzed by 13% SDS-PAGE and stained by Coomassie brilliant blue.

**Molecular dynamics simulation**. The crystal structure of the SHLD3-REV7-SHLD2 complex described in this manuscript was used for molecular dynamics simulations by removing SHLD2. Hydrogen atoms were added by SWISS PDB VIEWER. The system was solvated in a cuboid water box with $8.6 \times 7.5 \times 10.6$ Å$^3$ buffer and neutralizing counter ions were added. A concentration of 0.15 M NaCl salt bath was introduced to mimic experimental assay conditions. We used the OPLS-AA/L all-atom force field (2001 amino acid dihedrals) parameter set for the protein, and TIP3P model for water. Simulations were performed with GRO-MACS[41]. Prepared systems were first minimized using 5000 steps of a steepest descent algorithm, then equilibrated as follows: the system was heated to 310 K by Nose-Hoover in the isothermal–isobaric ($NpT$) ensemble over 25 ps. Production runs were then made for 120 ns duration in the $NpT$ ensemble. The short-range electrostatic and Lennard–Jones interactions were calculated within a cut-off of 12 Å. Particle Mesh Ewald was used for long-range electrostatics. Trajectory analysis was conducted with GROMACS. Before processing, the trajectories were aligned to calculate the RMSD between backbone atoms of the initial equilibrated structure and all subsequent frames.

**NHEJ assay**. Linearized DNA containing the EF1α promoter, the open reading frame of puromycin resistance and the WPRE element was expanded by PCR, using pCDH-CMV-MCS-EF1-Puro (System Biosciences) as template. Indicated cells were transfected with this linearized DNA and the pEGFP-C1 plasmid. Sixty hours later, the cells were collected and flow cytometry analysis was used to detect the efficiency of EGFP expression thus to determine the transfection efficiency. After incubation with medium containing puromycin for 14 days, the cells were fixed by 70% ethanol and stained with 0.1% Coomassie Brilliant Blue for 30 min at room temperature. The stained dishes were washed with water, and the colonies were counted by ImageJ (version 1.52a).

**Flag Pulldown**. Cells were lysed in NP-40 lysis buffer (50 mM Tris-HCl pH 7.5, 150 mM NaCl, 2 mM EDTA, 0.5% NP-50, 10 mM NaF, 1 mM PMSF and 1x cocktail protease inhibitor (Roche)). Lysates were then incubated with 10 μl anti-Flag agarose slurry at 4 °C for 4 hours. Anti-Flag agaroses were washed 3 times with NP-40 lysis buffer and boiled with 2× SDS-PAGE loading buffer. Protein concentrations were measured with the BCA protein assay (Thermo Scientific Pierce).

**S-tag Pulldown**. Cells were lysed in NP-40 lysis buffer (50 mM Tris-HCl pH 7.5, 150 mM NaCl, 2 mM EDTA, 0.5% NP-50, 10 mM NaF, 1 mM PMSF and 1× cocktail protease inhibitor (Roche)). Lysates were then incubated with 30 μl S-protein agarose slurry at 4 °C for 3 h. S-protein beads were washed 3 times with NP-40 lysis buffer and boiled with 2× SDS-PAGE loading buffer. Protein concentrations were measured with the BCA protein assay (Thermo Scientific Pierce). Uncropped western blots can be found in Supplementary Fig. 9.

**MBP Pulldown**. 30 μg MBP-SHLD2(1–60) proteins were first incubated with 20 μl dextrin beads slurry in buffer containing 20 mM Tris, pH 8.0, 150 mM NaCl, 1% glycerol and 0.5 mM TCEP at 4 °C for 3 h. Then dextrin beads were centrifuged and the supernatant was discarded. After that, 20 μg REV7-SHLD3(1–82) with or without 20 μg REV7 were added and incubated for another 30 min. Dextrin beads were washed 3 times with buffer containing 20 mM Tris, pH 8.0, 500 mM NaCl, 1% glycerol, 0.5 mM TCEP and 0.5% NP-40 and boiled with 2× SDS-PAGE loading buffer.

**Laser micro-irradiation and imaging of live cells**. Thirty-six hours after transfected with indicated plasmids, cells were plated on glass-bottomed dishes (NEST #801002) and sensitized with 5 μM Hoechst 33342 prior to exposure to a 405 nm localized laser beam (100% laser power, 24 s) on an inverted Nikon A1R microscope. Following micro-irradiation, cells were subject to live cell imaging.

**Quantification and statistical analysis**. Data were tested for statistical significance with GraphPad Prism 5 (Version 5.01). The tests performed, the number of biologically independent replicates ($n$) for each experiment are indicated in the Figure legends.

**Reporting Summary**. Further information on research design is available in the Nature Research Reporting Summary linked to this article.

## Data availability

Coordinates and structure factors for the crystal structure of human SHLD3-C-REV7-O-REV7-SHLD2 complex have been deposited into the Protein Data Bank with the accession code 6KTO. Structural details about REV7-REV3-REV1 (PDB ID: 3VU7) and C-Mad2-O-Mad2 (PDB ID: 2V64) are accessible in the Protein Data Bank (PDB). Source data for Figs. 2d–g, 3c–e, 4b–g, 5a–e, 7b–f, 8a–e and Supplementary Figs. 4a, b, 5a, b, 7d–f, 8a–c are provided as a Source Data file. All other data that support the study are available from the corresponding authors upon reasonable request.

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

## Acknowledgements

We thank the staffs from BL18U1 beamline of National Center for Protein Sciences Shanghai (NCPSS) at Shanghai Synchrotron Radiation Facility for assistance in data collection. We thank (Ms. Ping Wu, Mr. Yuxiang Sun from) Imaging Facility of National Center for Protein Sciences Beijing (NCPSB) and Lei Li for their assistance with Microscopy Imaging. We thank Prof. Jiadong Wang for help with NHEJ assay, Prof. Cai-Hong Yun, Zhucheng Chen, and Youdong Mao for critical discussion and Dr. Xiao-E Yan for advice on figure preparation. This work was supported by grants including the National Key Research and Development Program of China (Grant 2016YFA0500302 to Y.Y.), National Natural Science Foundation of China (Key grants 81430056, 31420103905, and 81621063 to Y.Y., grant 31800626 to L.L.), Beijing Natural Science Foundation (Key grant 7161007 to Y.Y.), and Lam Chung Nin Foundation for Systems Biomedicine. This work is also supported by High-performance Computing Platform of Peking University.

## Author contributions

L.L. and Y.Y. conceived the project. L.L. made the constructs, purified proteins, and performed crystallographic and biochemical experiments. J.F. performed cell assays. P.Z. and J.Y. performed cloning and purification of some mutants. Y.L. performed molecular dynamics simulation. L.L. solved the structure. L.L., J.F. and Y.Y. analyzed data and wrote the paper.

## Competing interests

The authors declare no competing interests.
