## [Peer Review File · Nature Communications]

Reviewers' comments:

Reviewer #1 (Remarks to the Author):

The work done by Liang et al. described the crystal structure of the Shieldin complex, and investigated its function in NHEJ. While this study is overall interesting, the results they presented are lack of clarity.

- 1) Whereas the quality of the electron density of the whole molecule is good, the local density at regions that are important for the formation of complex should be shown. Otherwise, it is hard to adjust the accuracy of the structure.
- 2) Fig.1b should be expanded with secondary structures of the O-REV7 labeled, while Fig. 1f seems meaningless, as the proposed dimer of tetramer is a minor population in solution, as shown by Fig. S2b.
- 3) It is more problematic that the assembly status of the samples of REV7WT-SHLD3 are quite heterogenous, as shown in Fig. S3b. This is also true for Fig. S2b. This issue is not discussed until Fig. 5a, in which the target complex turns out not to be a major species in solution. The mixture nature of the samples should be discussed and clarified, and the accuracy of the molecular weights measured by MALS should be indicated.
- 4) The colors for the positive- and negative charges in Fig. 2b are simply reversed.
- 5) It is interesting that the authors found REV7E35A-SHLD3(1-82) or REV7K44A-SHLD3(1-82), but not REV7K129A-SHLD3(1-82) or REV7K190A-SHLD3(1-82), bind to REV7K129A (O-REV7). Yet, the reasoning for these observation is not clearly stated, as the data and the discussion on the asymmetry are blended with other data.
- 6) The authors discuss quite extensively that SHLD3(1-37) contributes limited impact to the formation of C-REV7-O-REV7 dimer. Yet, no figure is used to show the structure of SHLD3(1-37). An expanded Fig. 3a to include the whole SHLD3 molecule may be suitable for this end.

Reviewer #2 (Remarks to the Author):

Cells are tribulated with endogenous and exogenous genotoxins that result in damaged-DNA. These damages are mainly repaired by two major pathways: 1) non-homologous end-joining (NHEJ) and homologous recombination (HR). The pathway choice for repair is considered one of the key factors that determines the fate of the cells. Moreover, there is increasing interest in some advanced cancer drugs (e.g. PARP inhibitors) that target defects in DNA repair pathways (e.g. HR) in cells. One resistance mechanism for PARP inhibitors in BRCA1-deficient cells is the loss of 53BP1-RIF1-REV7-shieldin pathway that downregulates NHEJ as shielding complex protects DNA breaks from resection. Also, shieldin proteins interact with translesion synthesis complex (involving REV1), which may provide additional resistance routes for cancer cells to escape the treatment. Thus, the ternary crystal structure of components (REV7-SHLD2-SHLD3) of the shielding complex described by Lian et al is timely and of broad interest for understanding resistance and potential therapeutic opportunities in the DNA damage repair-based cancer therapy.

The crystal structure presented here reveals the molecular assembly of the shieldin complex. Interestingly, the structure captures C (closed)-REV7-O (open)-REV7 dimer. This is stabilized by SHLD3 that wraps across the dimer using its REV7 binding motifs (RBMs). The authors found that SHLD2 facilitates the interaction between O-REV7 and the N-terminus of SHLD3 by forming a beta sheet insertion between them. The authors show that the conformational dimer seen in the crystal structure is critical for an efficient NHEJ activity. By overlaying the current structure with previously published REV3-C-REV7-REV1 structure, the authors find that the SHLD3 binding inhibits REV1 polymerase from binding to REV7, thus revealing exclusivity of NHEJ and REV1-mediated TLS. Furthermore, they have shown that the SHLD3-mediated REV7 dimer could bind to REV3 and may act as protein binding platform. Although the overall resolution of the crystal

structure (3.5 Å) is moderate, the authors have validated key interfaces by diligently making mutations and studying their impact on protein-protein interaction by ITC, MALS and gel-filtration assays. They also complemented their biophysical observation with NHEJ functional assays and cell measurements. The authors should address the following points.

Points.

- 1) The dimer of tetramer seen in the crystal structure (Figure 1f) and MALS (Supplementary Figure 2b) may be a concentration dependent assembly that may then only exist at higher concentrations. If this is a biologically relevant assembly making mutations at the interface should have an effect (e.g. NHEJ efficiency?). It would be helpful to test this with appropriate mutations.
- 2) Is the heterogeneity in the molecular weight observed in MALS data (Supplementary Figure 3b) for REV7wt-SHLD3 (1-82) because SHLD2 is missing from the experiment? As there is indication from MD simulation showing SHLD2 may help form a more stable complex, what happens to the molecular weight distribution (WT and mutants in supplementary figure 3) when you add SHLD2?
- 3) Fix typo line 164- FXPWFP instead of PXPWFP.
- 4) For completion, in Figure 4, could SHLD3 (when not bound to REV7) by itself interact with REV1?
- 5) While it is admirable to use gel-filtration assay to see the formation of complexes between different components, given the low resolution of Superdex200 column (in present case), it is hard to interpret the results. Ideally, these could be complemented with in vitro pulldowns using MBP tag on the SHLD2 (e.g. Figure 5)? It's just a suggestion.
- 6) The line 268 should read ...Interestingly, the structural alignment (Figure 6c) also shows beta1 of SHLD2 occupies the position of RBM2 of SHLD3 (as this is concluded from structural alignment of C-REV7-SHLD3 and O-REV7-SHLD2).
- 7) There were several different proteins and mutants expressed as part of this manuscript, were all of them purified using the same methods and buffer given in the general protein expression section of the methods? Addressing this would help with future reproducibility issues.
- 8) Were ITC experiments repeated? One would like to see a standard deviation for the measured binding affinity values.

Point-by-point Responses

"Molecular basis for assembly of the shieldin complex and its implications for NHEJ"
(NCOMMS-19-1124676-T)

Point-by-point responses to the comments by Reviewer #1:

The work done by Liang et al. described the crystal structure of the Shieldin complex, and investigated its function in NHEJ. While this study is overall interesting, the results they presented are lack of clarity.

1) Whereas the quality of the electron density of the whole molecule is good, the local density at regions that are important for the formation of complex should be shown. Otherwise, it is hard to adjust the accuracy of the structure.

Based on the reviewer's suggestion, we have included the local density at regions that are important for the formation of the complex in Supplementary Fig. 1, which is also shown below (Figure R1).

Figure R1 (Revised Supplementary Figure 1) Overview of the 2Fo-Fc electron density map of the SHLD3-REV7-SHLD2 complex and local density at regions important for the formation of the complex. The superimposed magenta, dark

yellow, light yellow and blue C α chains are that of SHLD3, C-REV7, O-REV7 and SHLD2, respectively.

a, Overview of the 2Fo-Fc electron density map of the SHLD3-REV7-SHLD2 complex. The map was contoured at 1.5 σ . **b-e**, Sections of the 2Fo-Fc electron density map surrounding some important regions contoured at 1.3 σ allows unambiguous placement of protein sidechains. RBM₂ of SHLD3 (**b**). The FXPWFP motif of SHLD3 (**c**). Some important residues at the conformational dimer interface (**d**). The interface between O-REV7 and SHLD2 centered around Tyr63 of O-REV7 (**e**). C-REV7 residues were shown with the carbon bonds colored dark yellow, while light yellow, red and blue carbon bonds indicate that of O-REV7, SHLD3 and SHLD2. Red and blue bonds represent that of oxygen and nitrogen.

2) Fig.1b should be expanded with secondary structures of the O-REV7 labeled, while Fig. 1f seems meaningless, as the proposed dimer of tetramer is a minor population in solution, as shown by Fig. S2b.

As the reviewer suggested, secondary structures of the O-REV7 were labeled in the revised Fig. 1b (Fig. R2), and we also deleted Fig. 1f and Fig. S2b as the reviewer pointed out that the proposed dimer of tetramer is a minor population in solution.

Figure R2 (Revised Figure 1b) Structure of the

SHLD3-C-REV7-O-REV7-SHLD2 complex. Two REV7 molecules are differentially colored to indicate their different states, C-REV7 is shown in green and O-REV7 is shown in cyan. The secondary structures of C-REV7, O-REV7, SHLD2 and SHLD3 are labelled. Disordered loop is shown as dashed lines.

3) It is more problematic that the assembly status of the samples of REV7^{WT}-SHLD3 are quite heterogeneous, as shown in Fig. S3b. This is also true for Fig. S2b. This issue is not discussed until Fig. 5a, in which the target complex turns out not to be a major species in solution. The mixture nature of the samples should be discussed and clarified, and the accuracy of the molecular weights measured by MALS should be indicated.

To clarify the mixture nature of the samples, we revised the manuscript as ‘Size-exclusion chromatography with multi-angle light scattering (SEC-MALS) shows the samples of REV7^{WT}-SHLD3 are quite heterogeneous and are proposed to be composed of C-REV7-O-REV7-SHLD3(1-82) and C-REV7-SHLD3(1-82), which were verified later (Supplementary Fig. 3b and Supplementary Table 1). Substitution of Glu35, Lys44, Arg124, Lys129 or Lys190 of REV7 at the dimer interface with an alanine residue abolishes formation of SHLD3 mediated REV7 conformational dimer and forms homogeneous C-REV7-SHLD3(1-82) heterodimer (Supplementary Fig. 3c-3h and Supplementary Table 1)’. In the revised figures, the accuracy of the molecular weights measured by MALS has been indicated by the fitted errors obtained from the data analysis software, which are showed in the brackets. We also calculated the error of observed masses compared with calculated masses (Table R1).

Table R1 (Revised Supplementary Table 1) The observed and calculated masses of the complexes.

	Observed Mass (kDa)	Calculated Mass (kDa)	Error (%)
REV7 ^{WT} -SHLD3(1-82)	55	36~62	ND
REV7 ^{R185A} -SHLD3(1-82)	57	36~62	ND
REV7 ^{E35A} -SHLD3(1-82)	37.1	36	3.1
REV7 ^{K44A} -SHLD3(1-82)	37.2	36	3.3
REV7 ^{R124A} -SHLD3(1-82)	34.1	36	-5.3
REV7 ^{K129A} -SHLD3(1-82)	35.3	36	-1.9
REV7 ^{K190A} -SHLD3(1-82)	34.7	36	-3.6
REV7 ^{Y63A} -SHLD3(1-82)	63	62	1.6
REV7 ^{W171A} -SHLD3(1-82)	65	62	4.8

4) *The colors for the positive- and negative charges in Fig. 2b are simply reversed.*

We are afraid that the reviewer has misunderstood Fig. 2b. We want to clarify that the residues shown in sticks on the surface are not residues of the electrostatic model. To avoid potential misunderstanding, we added following highlighted description in the revised figure legends. In left panel, C-REV7 is shown as electrostatic surface model and the labelled amino acid residues of O-REV7 that interact with C-REV7 are shown in sticks. In the right panel, O-REV7 is shown in electrostatic surface model while labelled residues of C-REV7 that interact with O-REV7 are shown in sticks.

5) *It is interesting that the authors found REV7E35A-SHLD3(1-82) or REV7K44A-SHLD3(1-82), but not REV7K129A-SHLD3(1-82) or REV7K190A-SHLD3(1-82), bind to REV7K129A (O-REV7). Yet, the reasoning for these observation is not clearly stated, as the data and the discussion on the asymmetry are blended with other data.*

We have added the following highlighted description in the revised version. This is because Lys129, Lys190 of C-REV7 locates at the asymmetric interface of C-REV7-O-REV7 and contributes to the interaction while Glu35, Lys44 of C-REV7 diverge from the interface and have no effect on the interaction.

6) *The authors discuss quite extensively that SHLD3(1-37) contributes limited impact to the formation of C-REV7-O-REV7 dimer. Yet, no figure is used to show the structure of SHLD3(1-37). An expanded Fig. 3a to include the whole SHLD3 molecule may be suitable for this end.*

As the reviewer suggested, we expanded Fig. 3a to include the whole SHLD3(1-37) to show its structure in the revised version (Fig. R3).

Figure R3 (Revised Figure 3a) The interface between SHLD3 and C-REV7-O-REV7. C-REV7 and O-REV7 are shown in electrostatic surface representation (positive potential, blue; negative potential, red), SHLD3 in ribbon view, and the FXPWFP motif in sticks. The N terminus of SHLD3 is also indicated. Residues 1-37 of SHLD3 are shown in ribbon.

Point-by-point responses to the comments by Reviewer #2:

Cells are tribulated with endogenous and exogenous genotoxins that result in damaged-DNA. These damages are mainly repaired by two major pathways: 1) non-homologous end-joining (NHEJ) and homologous recombination (HR). The pathway choice for repair is considered one of the key factors that determines the fate of the cells. Moreover, there is increasing interest in some advanced cancer drugs (e.g. PARP inhibitors) that target defects in DNA repair pathways (e.g. HR) in cells. One resistance mechanism for PARP inhibitors in BRCA1-deficient cells is the loss of 53BP1-RIF1-REV7-shieldin pathway that downregulates NHEJ as shieldin complex protects DNA breaks from resection. Also, shieldin proteins interact with translesion synthesis complex (involving REV1), which may provide additional resistance routes for cancer cells to escape the treatment. Thus, the ternary crystal structure of components (REV7-SHLD2-SHLD3) of the shieldin complex described by Liang et al is timely and of broad interest for understanding resistance and potential therapeutic opportunities in the DNA damage repair-based cancer therapy.

The crystal structure presented here reveals the molecular assembly of the shieldin complex. Interestingly, the structure captures C (closed)-REV7-O (open)-REV7 dimer.

This is stabilized by SHLD3 that wraps across the dimer using its REV7 binding motifs (RBMs). The authors found that SHLD2 facilitates the interaction between O-REV7 and the N-terminus of SHLD3 by forming a beta sheet insertion between them. The authors show that the conformational dimer seen in the crystal structure is critical for an efficient NHEJ activity. By overlaying the current structure with previously published REV3-C-REV7-REV1 structure, the authors find that the SHLD3 binding inhibits REV1 polymerase from binding to REV7, thus revealing exclusivity of NHEJ and REV1-mediated TLS. Furthermore, they have shown that the SHLD3-mediated REV7 dimer could bind to REV3 and may act as protein binding platform. Although the overall resolution of the crystal structure (3.5 Å) is moderate, the authors have validated key interfaces by diligently making mutations and studying their impact on protein-protein interaction by ITC, MALS and gel-filtration assays. They also complemented their biophysical observation with NHEJ functional assays and cell measurements. The authors should address the following points.

We would like to thank the reviewer for these positive comments. We have addressed each of the reviewer's comments as detailed below.

Points.

1) The dimer of tetramer seen in the crystal structure (Figure 1f) and MALS (Supplementary Figure 2b) may be a concentration dependent assembly that may then only exist at higher concentrations. If this is a biologically relevant assembly making mutations at the interface should have an effect (e.g. NHEJ efficiency?). It would be helpful to test this with appropriate mutations.

We tried to test this. However, it is difficult to make a mutation that only disturbs the dimer of tetramer while does not affect the C-REV7-O-REV7 conformational dimer. For example, Lys190 of O-REV7 locates at the interface of the dimer of tetramer and may contributes to the formation of the dimer of tetramer. However, REV7^{K190A} also fails to form the C-REV7-O-REV7 conformational dimer. On the other hand, as reviewer 1 suggested, the proposed dimer of tetramer is a minor population in solution, and thus Fig. 1f seems meaningless, so we deleted it in the revised version.

2) Is the heterogeneity in the molecular weight observed in MALS data (Supplementary Figure 3b) for REV7wt-SHLD3 (1-82) because SHLD2 is missing from the experiment? As there is indication from MD simulation showing SHLD2 may help form a more stable complex, what happens to the molecular weight distribution (WT and mutants in supplementary figure 3) when you add SHLD2?

Yes, our recombinant expressed REV7^{WT}-SHLD3(1-82) is not homogeneous and is composed of C-REV7-O-REV7-SHLD3(1-82) and C-REV7-SHLD3(1-82) without SHLD2. As shown in Fig. R4 (revised Fig. 5a), although excessive MBP-SHLD2(1-60) were added, some REV7^{WT}-SHLD3(1-82) samples form

MBP-SHLD2(1-60)-C-REV7-O-REV7-SHLD3(1-82) complexes, while others still exist as C-REV7-SHLD3(1-82). This can be confirmed by the observation that the ratio of REV7 to SHLD3(1-82) is higher in the peak eluted at 12.5~13 ml than in the peak eluted at 14.5~15 ml (Fig. R4, 9.6/1.0 versus 8.0/2.5, the gray values are labelled in black box in right panel). Meanwhile, since the mutant REV7^{R124A}-SHLD3(1-82) only exists as C-REV7-SHLD3(1-82) and fails to form a stable complex with MBP-SHLD2(1-60) in our gel-filtration assay (Fig. R4, red line in left panel and red box in right panel).

Figure R4 (Revised Figure 5a) SHLD3 mediated REV7 conformational dimer is essential for the recruitment of SHLD2. Gel filtration profiles show the interaction between REV7^{WT/R124A}-SHLD3(1-82) and MBP-SHLD2(1-60) in a Superdex200 Increase 10/300 size exclusion chromatography (SEC) column (the left panel). MBP-SHLD2(1-60) is excessive. The peaks eluted at 12.5~13 ml are composed of the stable complex of MBP-SHLD2(1-60)-REV7-SHLD3(1-82) while the peaks eluted at 14.5~15 ml are un-complexed MBP-SHLD2(1-60) or REV7-SHLD3(1-82). Numbers above bands represent the gray values of bands compared with SHLD3(1-82) eluted at 12.5~13 ml. Fractions (0.5 ml each) corresponding to REV7^{WT}-SHLD3(1-82) and MBP-SHLD2(1-60) co-elution (black), REV7^{R124A}-SHLD3(1-82) and MBP-SHLD2(1-60) co-elution (red) were analyzed by SDS-PAGE and stained by Coomassie brilliant blue (the right panel).

3) Fix typo line 164- FXPWFP instead of PXPWFP.

We apologize for this typo. We have corrected it.

4) For completion, in Figure 4, could SHLD3 (when not bound to REV7) by itself interact with REV1?

We performed ITC experiments which showed that SHLD3 by itself could not interact with REV1 (Figure R5). In Figure 4, we want to emphasize that REV1 interacts with C-REV7 (Fig. 4c and 4d) and the FXPWFP motif of SHLD3 disrupts

the interaction between REV1 and C-REV7 (Fig. 4b versus Fig. 4c and Fig. 4e versus Fig. 4f). To make it easier for interpreting these results, we have revised the description of Fig. 4 in the results section.

Figure R5 ITC measurement of the interaction between SHLD3(1-82) and cREV1 CTD. This is a representative result. N.D.: no detectable binding.

5) While it is admirable to use gel-filtration assay to see the formation of complexes between different components, given the low resolution of Superdex200 column (in present case), it is hard to interpret the results. Ideally, these could be complemented with *in vitro* pulldowns using MBP tag on the SHLD2 (e.g. Figure 5)? It's just a suggestion.

We thank the reviewer for pointing this out. As suggested, we did pull-down experiments using MBP tag on the SHLD2. In consistent with the gel-filtration assays, MBP-SHLD2(1-60) interacts with REV7^{WT}-SHLD3(1-82) (Fig. 5b, lane 3), but hardly interacts with the mutant REV7^{R124A}-SHLD3(1-82) that fails to form the conformational dimer (Fig. 5b, lane 4). Other mutants that fails to form the conformational dimer also show only weak interactions with MBP-SHLD2(1-60) in the pull-down assay (Fig. 5d, lane 9-12). In contrast, in presence of REV7^{K129A}, MBP-SHLD2(1-60) shows stronger interaction with REV7^{E35A/K44A}-SHLD3(1-82) that form reconstituted conformational dimer utilizing asymmetric REV7^{E35A/K44A} and REV7^{K129A} mutations than with REV7^{K129A/K190A}-SHLD3(1-82) that fails to form the conformational dimer with REV7^{K129A} (Fig. 5d, lane 13 and 14 versus lane 15 and 16). Moreover, to make it easier for interpreting the results, we marked the molecular weight and proteins corresponding to the peaks in the gel-filtration figures (Fig. 5a and 5c, both in the left panels).

Figure R5 (Revised Figure 5 a-d) SHLD3 mediated REV7 conformational dimer is essential for the recruitment of SHLD2.

a. Gel filtration profiles show the interaction between REV7^{WT/R124A}-SHLD3(1-82) and MBP-SHLD2(1-60) in a Superdex200 Increase 10/300 size exclusion chromatography (SEC) column (the left panel). MBP-SHLD2(1-60) is excessive. As indicated in the figure, the peaks eluted at 12.5~13 ml are composed of the stable complex of MBP-SHLD2(1-60)-REV7-SHLD3(1-82) while the peaks eluted at 14.5~15 ml are un-complexed MBP-SHLD2(1-60) or REV7-SHLD3(1-82). Numbers above bands represent the gray values of bands compared with SHLD3(1-82) eluted at 12.5~13 ml. Fractions (0.5 ml each) corresponding to REV7^{WT}-SHLD3(1-82) and MBP-SHLD2(1-60) co-elution (black), REV7^{R124A}-SHLD3(1-82) and MBP-SHLD2(1-60) co-elution (red) were analyzed by SDS-PAGE and stained by Coomassie brilliant blue (the right panel).

b. MBP pulldown showing the interaction between MBP-SHLD2(1-60) and REV7^{WT/R124A}-SHLD3(1-82). Samples were analyzed by SDS-PAGE and stained by Coomassie brilliant blue.

c. Gel filtration profiles show the interaction between REV7 mutants and MBP-SHLD2(1-60). Various mutants of REV7-SHLD3(1-82) complex (200 μ g) were first incubated with or without REV7^{K129A} (short as K129A in the Figure) (200 μ g), after 10 min, excessive MBP-SHLD2(1-60)

was added. Fractions (0.5 ml each) corresponding to REV7^{E35A}-SHLD3 927 (1-82)-REV7^{K129A} (short as E35A-1-82+K129A in the Figure) and MBP-SHLD2(1-60) co-elution (black), REV7^{K44A}-SHLD3(1-82)-REV7^{K129A} (short as K44A-1-82+K129A in the Figure) and MBP-SHLD2(1-60) co-elution (red), REV7^{K129A}-SHLD3(1-82)-REV7^{K129A} (short as K129A-1-82+K129A in the Figure) and MBP-SHLD2(1-60) co-elution (blue), REV7^{E35A}-SHLD3(1-82) (short as E35A-1-82 in the Figure) and MBP-SHLD2(1-60) co-elution (pink) were analyzed by SDS-PAGE and stained by Coomassie brilliant blue (the right panel).

d. MBP pulldown assay shows the interaction between MBP-SHLD2(1-60) and REV7-SHLD3(1-82) mutants or reconstituted conformational dimer. Samples were analyzed by SDS-PAGE and stained by Coomassie brilliant blue.

6) The line 268 should read ...Interestingly, the structural alignment (Figure 6c) also shows beta1 of SHLD2 occupies the position of RBM2 of SHLD3 (as this is concluded from structural alignment of C-REV7-SHLD3 and O-REV7-SHLD2).

We have revised it as the reviewer suggested.

7) There were several different proteins and mutants expressed as part of this manuscript, were all of them purified using the same methods and buffer given in the general protein expression section of the methods? Addressing this would help with future reproducibility issues.

Yes, all the proteins used in the manuscript were purified using the same methods and buffer given in the general protein expression section of the methods. To address this, we added 'All proteins used in this study were purified using the same methods and buffer as described above' in the revised version.

8) Were ITC experiments repeated? One would like to see a standard deviation for the measured binding affinity values.

We showed representative results previously. Now we have calculated the standard deviations for the measured binding affinity values from three independent experiments in the revised version (revised Figs. 2d-g, 3c-e, 4b-g, 7c, 7e, 8a, 8c, and Supplementary Figs. 4a-b, 5a-b, 7d-e and 8b-c). The following Table R2 summarizes the results of repeated experiments.

Table R2 ITC experiment statistics

Syringe	Cell	K _D 1 (μM)	K _D 2 (μM)	K _D 3 (μM)	mean K _D (μM)	STDEVP
cREV1 CTD	REV7 ^{K44A} -SHLD3(1-82) ^{5A}	1.04	1.05	1.25	1.1	0.10
cREV1 CTD	REV7 ^{K44A} -SHLD3(45-82)	1.7	1.24	1.11	1.4	0.25
cREV1 CTD	REV7 ^{WT} -REV3(1871-2021)	1.4	1.3	2.2	1.6	0.40
cREV1 CTD	REV7 ^{K44A} -SHLD3(1-82) ^{F38A}	4.8	7.3	7.4	6.5	1.2
cREV1 CTD	REV7 ^{K44A} -SHLD3(38-82)	ND	ND	ND		
cREV1 CTD	REV7 ^{K44A} -SHLD3(1-82)	ND	ND	ND		
REV7 ^{K44A} -SHLD3(1-82) ^{F38A}	REV7 ^{K129A}	1.7	1.9	1.7	1.8	0.09
REV7 ^{K44A} -SHLD3(28-82)	REV7 ^{K129A}	0.21	0.2	0.18	0.2	0.01
REV7 ^{K44A} -SHLD3(38-82)	REV7 ^{K129A}	0.71	0.3	0.4	0.47	0.17
REV7 ^{K44A} -SHLD3(1-82)	REV7 ^{K129A}	0.09	0.11	0.12	0.11	0.01
REV7 ^{E35A} -SHLD3(1-82)	REV7 ^{K129A}	0.25	0.3	0.28	0.28	0.02
REV7 ^{K129A} -SHLD3(1-82)	REV7 ^{K129A}	ND	ND	ND		
REV7 ^{K190A} -SHLD3(1-82)	REV7 ^{K129A}	ND	ND	ND		
REV3(1847-1906)	REV7 ^{E35A} -SHLD3(1-82)-REV7 ^{K129A}	0.053	0.049	0.037	0.046	0.0068
REV7 ^{E35A} -SHLD3(1-82)	REV7 ^{K129A} -REV3(1847-1906)	4.3	5.6	9.9	6.6	2.4
MBP-SHLD2(1-60)	REV7 ^{W171A} -SHLD3(1-82)	0.044	0.05	0.083	0.059	0.017
MBP-SHLD2(1-60)	REV7 ^{Y63A} -SHLD3(1-82)	0.39	0.98	0.44	0.60	0.27
SHLD3(1-82)	REV7 ^{WT}	0.075	0.25	0.075	0.13	0.083
SHLD3(1-82)	REV7 ^{K129A}	0.0087	0.0029	0.0027	0.0048	0.0028
REV3(1847-1906)	REV7 ^{K129A}	0.0046	0.0084	0.0089	0.0073	0.0019
SHLD3(1-82)	REV7 ^{W171A}	0.19	0.16	0.16	0.17	0.014
SHLD3(1-82)	REV7 ^{Y63A}	0.17	0.21	0.1	0.16	0.046
REV7 ^{K44A} -SHLD3(1-82) ^{5A}	REV7 ^{K129A}	2.1	3	4.8	3.3	1.1
REV7 ^{K44A} -SHLD3(45-82)	REV7 ^{K129A}	4.2	3.8	4.2	4.1	0.19
REV7 ^{E35A} -SHLD3(1-82)	REV7 ^{K129A} +MBP-SHLD2(1-60)	0.041	0.054	0.085	0.060	0.019
REV7 ^{K44A} -SHLD3(1-82)	REV7 ^{K129A} +MBP-SHLD2(1-60)	0.62	0.27	0.35	0.41	0.15
MBP-REV3(1871-2021)	REV7 ^{W171A} -SHLD3(1-82)	0.55	0.89	1.74	1.06	0.50
MBP-REV3(1871-2021)	REV7 ^{Y63A} -SHLD3(1-82)	12	53	47	37	18

REVIEWERS' COMMENTS:

Reviewer #1 (Remarks to the Author):

I am satisfied with the responses.

Reviewer #2 (Remarks to the Author):

The authors did a thorough job on addressing both reviewers' comments. Furthermore the authors revisions in response to the reviewers' comments have improved the manuscript for publication.

Minor points.

Fix sentence ending "regulatory role in choice of DSB repair pathway at the earliest 10, 11" - i.e. Observed Timepoint or ?

Include standard deviations in the binding constant measured from three independent ITC experiments while discussing in the main text.

Point-by-point Responses

"Molecular basis for assembly of the shieldin complex and its implications for NHEJ"
(NCOMMS-19-1124676A)

Point-by-point responses to the comments by Reviewer #1:

I am satisfied with the responses.

We thank the reviewer for the comments and time.

Point-by-point responses to the comments by Reviewer #2:

The authors did a thorough job on addressing both reviewers' comments. Furthermore, the authors' revisions in response to the reviewers' comments have improved the manuscript for publication.

We thank the reviewer for the comments and time.

Minor points.

Fix sentence ending 'regulatory role in choice of DSB repair pathway at the earliest^{10, 11}, - i.e. Observed Time-point or?

Yes, it is observed time-point and we have revised this sentence.

Include standard deviations in the binding constant measured from three independent ITC experiments while discussing in the main text.

We have revised the main text as the reviewer suggested.